# Resolving forebrain developmental organisation by analysis of differential growth patterns

Elizabeth Manning [1,2,3,12], Kavitha Chinnaiya [1,2,3], Caitlyn Furley[1,2,3], Dong Won Kim[4,5], Seth Blackshaw [6,7,8,9,10,11], Marysia Placzek [1,2,3] & Elsie Place [1,2,3,12]

The forebrain is the most complex region of the vertebrate central nervous system, and its developmental organisation is controversial. We fate-mapped the embryonic chick anterior neural tube and built a 4D model of brain growth. We reveal modular patterns of anisotropic growth, ascribed to progenitor regions through multiplex hybridisation chain reaction. Morphogenesis is dominated by directional growth towards the eye, more isometric expansion of the prethalamus and dorsal telencephalon, and anterior movement of ventral cells into the hypothalamus. Comparative gene expression analysis and cell mixing experiments suggest the existence of a contiguous transverse boundary region, encompassing the *zona limitans intrathalamica* and retro-mammillary hypothalamus, that divides the anterior and posterior forebrain, and becomes distorted at the base of the *zona limitans intrathalamica*. Fate conversion experiments indicate that the hypothalamus is topologically tri-partite, lying ventral to the telencephalon, prethalamus and *zona limitans intrathalamica*. Our findings challenge the widely accepted prosomere model of forebrain organisation, do not support a segmented anterior forebrain, and instead suggest a 'tripartite hypothalamus' model.

In recent years, remarkable progress has been made in the profiling and classification of central nervous system cell types, and in understanding their specification and differentiation from neural progenitors[1–3]. Nonetheless, the fundamental layout of the developing vertebrate brain is still disputed, reflected in a confused and contradictory contemporary literature, with the extent and position of the hypothalamus at the centre of the debate. In classic columnar models of forebrain organisation, the thalamus, prethalamus, hypothalamus

and eye were held to belong to the diencephalon, with the telencephalon positioned anterior to these structures[4]. However, an alternative view, in the form of the 'prosomere' model, has since gained significant traction. In this scheme, the forebrain consists of seven transverse segments—'prosomeres'—analogous to the neuromeres of the hindbrain and spinal cord. Each prosomere possesses both roof plate and floor plate, a conceptual necessity to meet the prosomere definition of a segment. The hypothalamus is grouped with

[1]School of Biosciences, University of Sheffield, Sheffield, UK. [2]Bateson Centre, University of Sheffield, Sheffield, UK. [3]Neuroscience Institute, University of Sheffield, Sheffield, UK. [4]Danish Research Institute of Translational Neuroscience (DANDRITE), Nordic EMBL Partnership for Molecular Medicine, Aarhus University, Aarhus, Denmark. [5]Department of Biomedicine, Aarhus University, Aarhus, Denmark. [6]Solomon H. Snyder Department of Neuroscience, Johns Hopkins University School of Medicine, Baltimore, MD, USA. [7]Department of Psychiatry and Behavioral Science, Johns Hopkins University School of Medicine, Baltimore, MD, USA. [8]Department of Ophthalmology, Johns Hopkins University School of Medicine, Baltimore, MD, USA. [9]Department of Neurology, Johns Hopkins University School of Medicine, Baltimore, MD, USA. [10]Institute for Cell Engineering, Johns Hopkins University School of Medicine, Baltimore, MD, USA. [11]Kavli Neuroscience Discovery Institute, Johns Hopkins University School of Medicine, Baltimore, MD, USA. [12]These authors contributed equally: Elizabeth Manning, Elsie Place. ✉e-mail: m.placzek@sheffield.ac.uk; e.place@sheffield.ac.uk

the telencephalon, together occupying two anteriormost prosomeres (hp2, hp1). Posterior to these are three diencephalic prosomeres (p3-p1) housing the prethalamus (p3), thalamus (p2), pretectum (p1) and associated ventral structures[5,6]. Most recently the midbrain has been designated as forebrain, contributing two more prosomeres[7].

Although influential, the prosomere model is controversial, relying heavily for its support on gene expression patterns in the mid-to-late stage embryonic mouse. Classic features of segment boundaries, such as lineage restriction, have not been identified at any purported prosomere boundaries anterior to the ZLI (the prosomere dorsal p2/p3 boundary), nor have the locations of prosomere boundaries in the early ventral forebrain been precisely defined. And while the model incorporates detailed fate-mapping data—which can contribute valuable insight into topological relationships within the developing forebrain—this evidence is largely restricted to the dorsal aspect of the chicken neural tube, leaving ventral and basal regions under-characterized. A further challenge has arisen from recent studies showing that hypothalamic patterning is closely coordinated with prethalamic, rather than telencephalic, patterning[8-12].

Here we clarify forebrain developmental organisation. We completed the fate map for the Hamburger-Hamilton stage 10 (HH10) chicken anterior neural tube while systematically visualising growth patterns, enabling a 4D reconstruction of forebrain growth. DiI/DiO 'growth lines' reveal how region-specific patterns of anisotropic growth sculpt the amniote forebrain. We reveal progenitors that originate in close proximity, and therefore receive similar positional cues during critical patterning periods, but then move apart, their original topological relations obscured by differential and directional growth. Molecular studies of the early chicken and mouse forebrain, supported by fate conversion and cell mixing experiments, contradict key predictions of the prosomere model and suggest alternative models. Our findings indicate a central role of eye morphogenesis in separating the telencephalon from the prethalamus and hypothalamus, all within a common anterior forebrain subdivision. A contiguous transverse boundary region, encompassing the retromammillary hypothalamus and *zona limitans intrathalamica* (ZLI), divides the anterior and posterior forebrain; region-specific growth distorts this transverse morphology, creating a flexure in the D-V axis at the base of the ZLI. Our work resolves prior misinterpretation and provides a new model for forebrain developmental organisation that we term the 'tripartite hypothalamus' model.

## Results

### DiI/DiO 'growth lines' reveal forebrain anisotropic growth

We fate-mapped the chicken anterior neural tube at HH10, aided by prominent morphological landmarks including a series of transient epithelial folds in the ventral midline (Figs. 1A-C and S1A-D). Embryos were injected with DiI/DiO, targeting the twelve zones depicted in Fig. 1D, and analysed after 48 hours (HH17-20), with a subset analysed after 24 hours (HH14-15). We labelled relatively large numbers of cells to provide insight into tissue-level growth patterns, and describe the resulting DiI/DiO distributions on the basis of morphology and marker genes (Figs. 1E, and S1E)[8,13,14]. We use a standard definition of the hypothalamus, namely the paraventricular (PV), tuberal, mammillary (MM) and retromammillary (RM, also known as supramammillary) domains, omitting the preoptic region which is now usually considered part of the telencephalic subpallium. We define the A-P axis as running parallel to the hypothalamic *NKX2-2* expression domain and the D-V axis as orthogonal to it (Fig. 1F).

Labelled neuroectoderm reproducibly gave rise to distinctively shaped territories, referred to as 'growth lines' (Fig. 1G) that could be allocated to one or more of seventeen forebrain regions (Fig. 1H), together revealing highly anisotropic region-specific growth. Injection zones 7-9 give rise to growth lines that reproducibly stretch from the telencephalon to the dorsal optic stalk or optic midline (Figs. 1I-L and

S2)—zone 9 also skirting the prethalamic eminence (Fig. S2). Growth lines extend into the nasal retina where injections approached or overlapped with zone 10 (Fig. 1K).

Growth lines originating from ventral zones are elongated in an A-P direction (Figs. 1L-O and S3 and S4). Those originating from lateral parts of zones 4 and 5 (not previously fate-mapped) stretch from the *SHH*$^{+ve}$ hypothalamus, across the alar-basal boundary (ABB), into the ventral optic stalk and temporal retina, mirroring the growth lines connecting the telencephalon with nasal retina (Figs. 1L, M and S3). The long growth lines originating in zones 7-10 and 4-5 therefore converge towards the eye, and tend to lie within either *FOXG1*$^{+ve}$ or *FOXD1*$^{+ve}$ domains, respectively (Figs. 1P and S3R,S). They outline a wedge-shaped area of more isometric growth encompassing the prethalamus, which originates in area 11 (Figs. 1Q and S4C, D).

Zones 1 and 2 label the ventral hypothalamus. As previously demonstrated[13,14], cells closest to the midline remain near the ventral midline while maintaining their relative A-P positions (Figs. 1R, S and S2O,P). Wider midline injections form an anteroventrally-sloping V-shape, showing that midline cells are displaced anteriorly relative to their more lateral counterparts (Figs. 1T, yellow arrow; S3E, S4J-K; 1U). This pattern is also clearly seen in growth lines arising from zones 2/6, which slope anteroventrally towards the tuberal hypothalamic midline, curving around the end of the *SHH*$^{+ve}$ floor plate and becoming distorted as they cross the *PITX2*$^{+ve}$ RM hypothalamus (Figs. 1V and S3K,L). The anterior-most *SHH*$^{+ve}$ floor plate arises from zone 3 (Fig. 1W).

Zone 12 gives rise to growth lines in the anterior thalamus and pretectum, which align in a D-V direction (Fig. S2K, zone 12). Large/multiple injections that simultaneously target zones 2/6/12 show that the D-V growth lines from zone 12 are continuous with the A-P growth lines in the basal hypothalamus: the orientation of growth lines changes abruptly close to the ABB, just posterior to the ZLI (Figs. 1W, X and S4H,I). Zone 6 injections give rise to a small tricorn shape at this point (Fig. 1Z).

In summary, our data shows that the hypothalamus arises in zones 1, 2, 4 and 5 (Fig. 1A, full data in Fig. S5), and that morphogenesis of the forebrain occurs in a stereotyped pattern of anisotropic growth, which is manifest as a series of highly reproducible, overlapping growth lines (Figs. 1G, AB). Growth lines maintain their relative topology, with minimal to no mixing of dye observed from different injection sites (Figs. 1AA, AB, S2-S5, for example see Fig. S4K). Key growth patterns, including the V-shaped and A-P oriented lines in the hypothalamus, and D-V oriented lines in dorsal regions posterior to the ZLI, were confirmed by DiI-DiO labelling of HH7-9 neuroectoderm (Fig. S6), by genetic clonal analysis using the Cre-Cytbow transgenic line (Fig. S7A-C)[15], and in RFP-electroporated embryos (Fig. S7D-F).

### Forebrain morphogenesis through region-specific growth patterns

We built a four-dimensional model of forebrain growth, based on the above findings combined with existing fate maps for dorsal regions[16-18] (Fig. S8). This provides the most complete fate map yet of the HH10 chicken forebrain (Figs. 2A and S9, Movie S1) and recapitulates key growth patterns in 'digital DiI' simulations (Fig. 2B, Movie S2). In sum, the model describes the morphological transformation of the anterior neural tube to the characteristic form of the amniote forebrain.

Anterior to the ZLI, elongated growth lines converge towards the optic stalk, demonstrating that eye outgrowth is associated with strongly directional growth over a wide area of the forebrain (Fig. 2B, Movie S3 0:16-0:29). Up to HH20, anterior forebrain morphogenesis is dominated by the combination of this directional growth, and a more uniform expansion of the telencephalic pallium and the prethalamus (Fig. 2C, Movie S3 0:35-1.00). The telencephalon thus takes on a balloon-like shape, narrowing towards the optic stalk; the prethalamus and PV hypothalamus together become wedge-shaped, narrowing

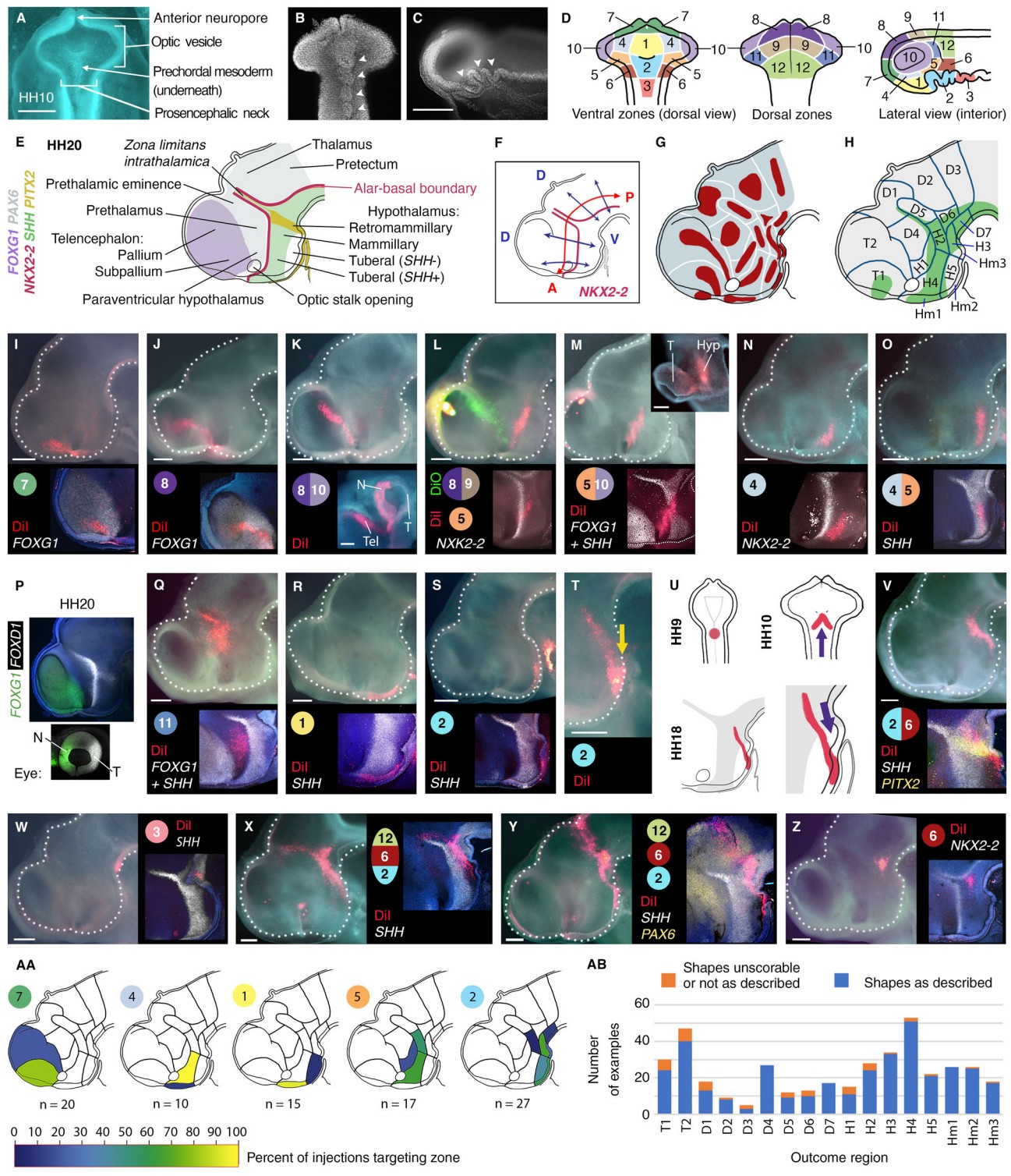

**Fig. 1 | Fate mapping reveals forebrain growth patterns. A-C** HH10 chick morphology. **A** In ovo dorsal view, dorsal neuroectoderm removed. **B** DAPI-stained neuroepithelium, ventral view (*n* > 50). **C** hemisected, internal view (*n* > 100). Arrowheads = midline folds. **D** HH10 injection zones. **E** HH20 forebrain regions and marker genes. **F** A-P/D-V axes. **G** Outcome regions used for classifying DiI/DiO results. **H** Characteristic 'growth line' shapes seen in each outcome region. **I-O, Q-T, V-Z** Hemisected HH17-20 heads showing 'growth lines', DiI/DiO injected at HH10, internal views (*n* numbers indicated in AA and AB). Circled numbers indicate HH10 injection zones in (**D**); split circles indicate zone intersection points; multiple circles indicate multiple injections. Upper image: brightfield plus DiI/DiO. Lower/right:

HCR in situ results for selected genes as indicated, same sample except in (**Q**); **K** lower panel and **M** upper inset - outer view of eye, lateral (**K**) and posterior (**M**) views, flipped horizontally to aid interpretation. **P** HCR-processed internal and outer eye view, HH20 (*n* = 5). **U** Posterior ventral midline injection at HH9 forms V-shapes at HH10 and HH18. AA) Outcome regions labelled following targeting of zones 7, 4, 1, 5 and 2; white = no examples recorded. See Supplementary Fig. S5 for full results. AB) Quantitative analysis of growth line shapes in outcome regions. Hyp hypothalamus, N Nasal eye, T Temporal eye. Scale bars − 250 μm. Source data for (AA) and (AB) are provided as a Source Data file.

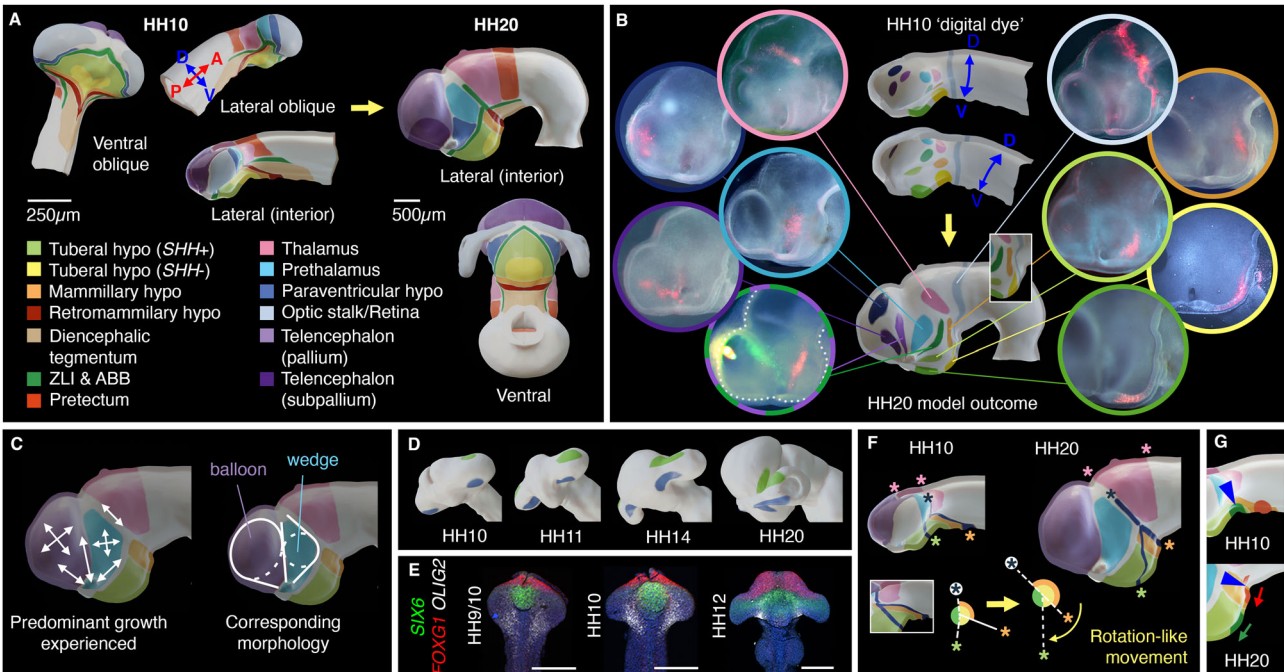

**Fig. 2 | 4-D model of forebrain growth. A** Stills from 4-D reconstruction of HH10-HH20 fate maps. Colours show prospective forebrain regions, and their topological relationships, at HH10 and HH20. **B** 'Digital dye' applied at HH10 (ventrolateral and dorsolateral views shown to aid visualisation) with resulting growth patterns, and comparison to in vivo examples. **C** Simplified growth patterns (upper), and morphology resulting from isometric (circles) and directional (lines) growth (lower). **D** 'Digital dye' spots progress to growth lines connecting eye and forebrain; oblique anterior-ventral views. **E** HCR-stained neuroectoderms, ventral views (*n* = 3-4 per stage). **F** Model depicted as simplified fate map with dark lines and asterisks tracking position of thalamus and posterior hypothalamic cells relative to the ZLI. Lines intersect at a 'hinge'-like point. **G** Movement of 'digital dye' spots (green, red; arrows) in relation to HH10 ventral inflection point (upper, asterisk) and HH20 cephalic flexure (lower, asterisk). Hypo hypothalamus, ABB alar-basal boundary, ZLI zona limitans intrathalamica. Scale bars − 250 μm.

towards the dorsal midline and optic stalk (Fig. 2C); the optic vesicle/cup/stalk opening holds to a progressively smaller corner of the anterior forebrain (Fig. S10A-D, Movie S3 1:03-1:20). Telencephalic and basal hypothalamic lineages are largely segregated from an early stage by the intervening prethalamus, PV hypothalamus and eye field (Movie S3 1:22-1:47; compare growth lines in Fig. 2B to HH20 regions in Fig. 2A). Growth lines connect regions sharing either *FOXG1* (telencephalon and nasal retina) or *SIX6/FOXD1* (hypothalamus and temporal retina) expression (Figs. 2D, E, 1K, S10E, Movie S3 1:48-2:30).

More posteriorly, the thalamus also expands, contributing to the dorsal curvature of the neural tube (Fig. 2F, pink asterisks; Movie S4 0:08-0:18). Cells ventral to the ABB are displaced anteriorly by extension of the ventral midline, bringing them towards the telencephalon and creating V-shaped growth lines (Fig. 2F, green/orange asterisks; Movie S4 0:35-1:35). This movement resembles an anterior rotation of ventral regions, with a 'hinge' point at the intersection of the ABB and ZLI (Fig. 2F; Movie S4 1:34-1:48). The epithelial folds in the HH10 ventral midline contribute to the ventral tuberal hypothalamus; the ventral inflection point of the HH10 neural tube does not correspond to the HH20 cephalic flexure (Fig. 2G, asterisks; Movie S4 1:51-end), as is sometimes assumed.

**The dorsoventral axis becomes distorted around the ZLI base**

Proponents of the prosomere model argue that higher proliferation in the dorsal neural tube[17,19] distorts block-shaped proto-segments into wedge-shaped prosomeres, causing the neural tube to pivot around the cephalic flexure (Fig. 3A)[20–23]. This provides the explanation for the position of the floor plate and roof plate, and hence the rationale for the orientation of the A-P and D-V axes (Fig. 3B), and the resulting placement of the hypothalamus ventral to the telencephalon and anterior to the prethalamus (Fig. 3C). However, when purported

prosomere boundaries, based on earlier fate mapping[16], are drawn onto our model at HH10 and HH20 (Fig. 3D top panels), their positions are not as predicted when viewed at HH20 and HH10, respectively (Fig. 3D bottom panels). Instead, our fate-mapping shows that much of the ventral hypothalamus arises posterior to the telencephalon, and ventral to areas that will give rise to the PV hypothalamus, prethalamus and thalamus (Figs. 2A,F and 3D, E). The anterior tuberal hypothalamus is an exception, originating ventral to the telencephalon.

We hypothesised that the early proximity of prospective hypothalamic and diencephalic cells results in a similar positional identity, due to similar exposure to fate-determining cues. If this is the case, it should be reflected in the shapes of progenitor domains towards the end of patterning stages (by -HH20). Intriguingly, several genes in the HH20 posterior hypothalamus run approximately parallel to the ZLI, a lineage-restricted dorsal diencephalic signalling centre that divides the vertebrate nervous system into anterior *SIX/FEZ*⁺ᵛᵉ and posterior *IRX*⁺ᵛᵉ regions[24–26]. These include *PITX2* and *LMX1B* in the RM hypothalamus, *EMX2, OLIG2*[8], *EPHA7*, and *FOXD1* in the MM hypothalamus, and *LHX6* and *ARX* in the prethalamic-like tuberomammillary terminal (TT) (Figs. 3F-J; 1E,P; S1E; S11A-F and ref. [8]). Notably, *PITX2* overlaps with a distinct *FOXA1/LMX1B/SHH*⁺ᵛᵉ ventral midline domain that abuts the *ARX*⁺ᵛᵉ floor plate[27] at a flexure point (Fig. 3H, arrowhead; S11B,G). The *PITX2*⁺ᵛᵉ RM is flanked by, and partly overlaps with, *WNT8B, SIM1, EPHA7* and *DBX1*; these genes converge at the ZLI and ventral midline (Figs. 3I; S11C-E), *WNT8B* resolving from a wider uniform domain at earlier stages (Fig. S11H).

Strikingly, *WNT8B* and *BARHL2* encompass the ZLI dorsally and extend continuously through the hypothalamus to the *PITX2/SHH*⁺ᵛᵉ ventral midline (Figs. 3I, J and S11I-J). A slender, straight *LFNG*⁺ᵛᵉ domain encompasses the ZLI[28], *PITX2*⁺ᵛᵉ SM, and part of the hypothalamic *WNT8B/BARHL2*⁺ᵛᵉ domain (Figs. 3I, J and S11I-K). The *PITX2*⁺ᵛᵉ RM is

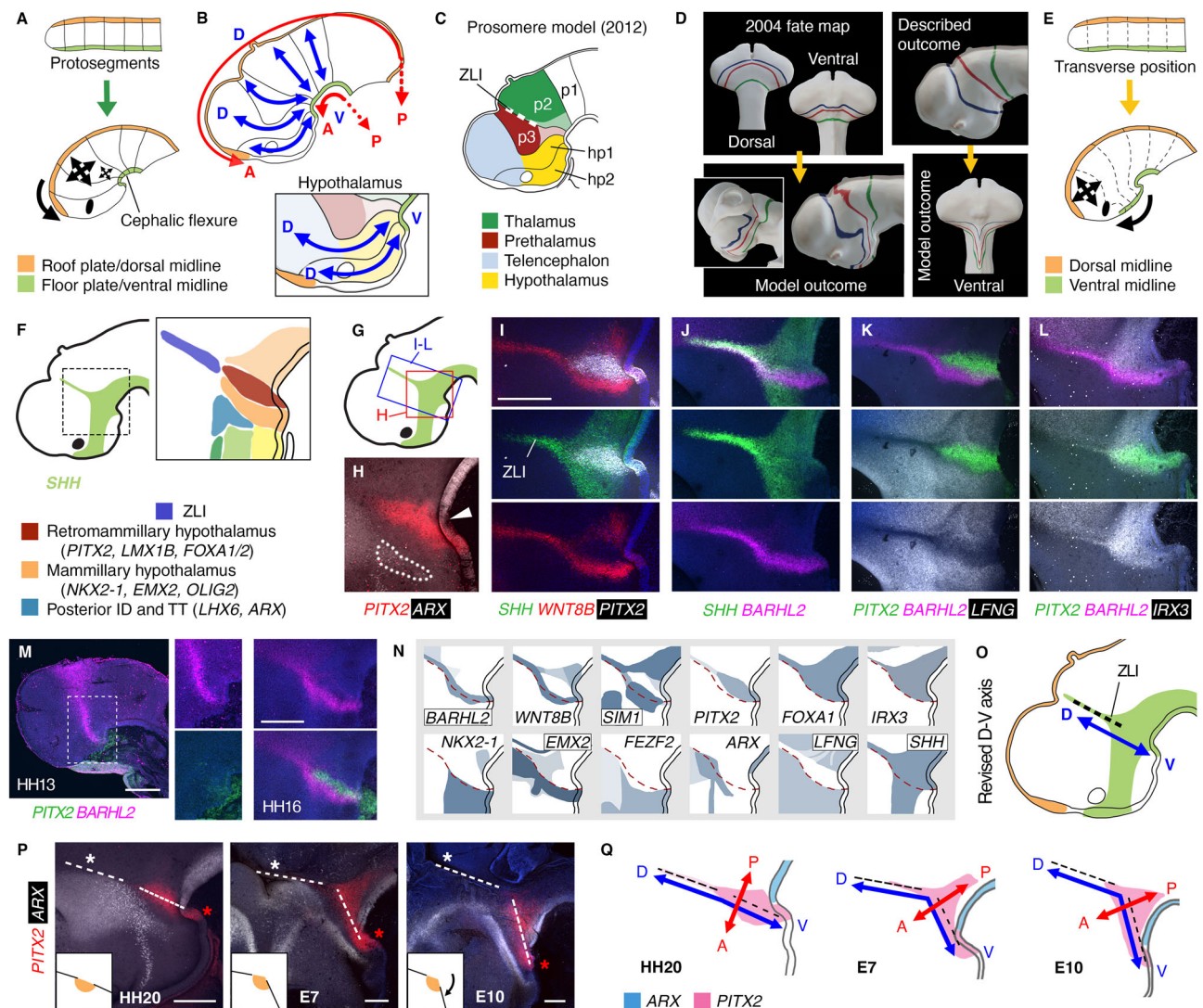

**Fig. 3 | Repositioning of the forebrain dorsoventral axis. A** Growth model underlying prosomere model. Transverse lines represent incipient interprosomeric boundaries. Two-headed arrows–differential growth, dorsal versus ventral regions. Curved arrow–neural tube bending. **B** Prosomere model axes. **C** 2012 prosomere model segments. P1-3–diencephalic prosomeres 1-3, hp1-2 - hypothalamo-telencephalic prosomeres 1-2[6,7]. **D** 2004 prosomere model boundaries at HH10 as per[16] (top left), and at HH20 (top right). 4D model outcome (gold arrows) of the same prosomere lines at HH20 (bottom left) and HH10 (bottom right). **E** Revised growth model. Dotted lines represent cells at particular transverse positions at the earlier stage, and their descendants' positions after neural tube bending. Two-headed arrows - telencephalic expansion. Curved arrow - movement of ventral midline cells. **F** Schematic of ZLI and posterior hypothalamic regions. **G** Location of regions shown in **H-L. H-M**) HCR analysis of ZLI and posterior hypothalamic genes (*n* = 3-30 per probe per stage). **H** Arrowhead–flexure where *ARX*[+ve] floor plate abuts *PITX2*[+ve] ventral midline. Dotted outline–*ARX* in hypothalamic tuberomammillary terminal[11]. **N** Selected HH20 gene expression patterns, posterior hypothalamus and surroundings. Dotted line–edge of *LFNG*[+ve]/*IRX3*[+ve] area. **O** Revised dorsoventral axis. **P** HCR, HH20-E10 hypothalamus and ZLI. Dotted lines through the ZLI (white asterisks) and *PITX2* RM region meet at an angle, resembling a 'hinge' point (inserts). Red asterisks–*PITX2*[+ve] ventral midline (*n* ≥ 3 per stage). **Q** Distortion of the A-P and D-V axes between HH20 and E10. Scale bars – 250 μm. A anterior, P posterior, D dorsal, V ventral, ID Intrahypothalamic Diagonal, TT Tuberomammillary Terminal, ZLI zona limitans intrathalamica.

*IRX1/IRX3*[+ve] and *FEZF2*[+ve], as in mouse[9,29,30] (Figs. 3L, and S11L-M), i.e. has a 'posterior' molecular profile. *BARHL2* expression straddles the edge of *IRX3* in the RM and ZLI (a slight kink is less pronounced at earlier stages, prior to the onset of *PITX2* expression)(Figs. 3L, M and S11L). In keeping with the posterior origin of these cells (Fig. 2A), this 'posterior' molecular profile places the RM hypothalamus both posterior to the prethalamus and contiguous with the ZLI (selected expression patterns are summarised in Fig. 3N). The D-V axis within the hypothalamus is therefore rotated compared to the prosomere model, in which the prethalamus is posterior to the RM hypothalamus (Fig. 3C).

To investigate whether continued growth may disguise these topological relations, we examined selected gene expression patterns

at later developmental stages. At E7-E10 many genes including *PITX2*, *ARX*, *WNT8B*, *LFNG*, *EPHA7*, *FOXA1*, *OTP*, *SIM1* and *FOXA2* bear similar relations to one another as at HH20 (Figs. 3P and S12A-E). However, recalling the anterior movement of ventral cells at earlier stages (Fig. 2), the *PITX2*[+ve] ventral midline has shifted anteriorly relative to the ZLI. This creates an obtuse angle between the SM domain and the ZLI, recalling the 'hinge point' in our growth model (Figs. 3P insets; 2F). The D-V/transverse alignment of these regions becomes distorted and the MM/RM domains start to take on an A-P oriented/longitudinal appearance (Fig. 3Q), the anterior edge of *WNT8B* aligning with a characteristic morphological fold, the hypothalamic periventricular organ (Fig. S12F)[31,32]. The topology of the chick embryonic forebrain thus becomes obscured by differential growth.

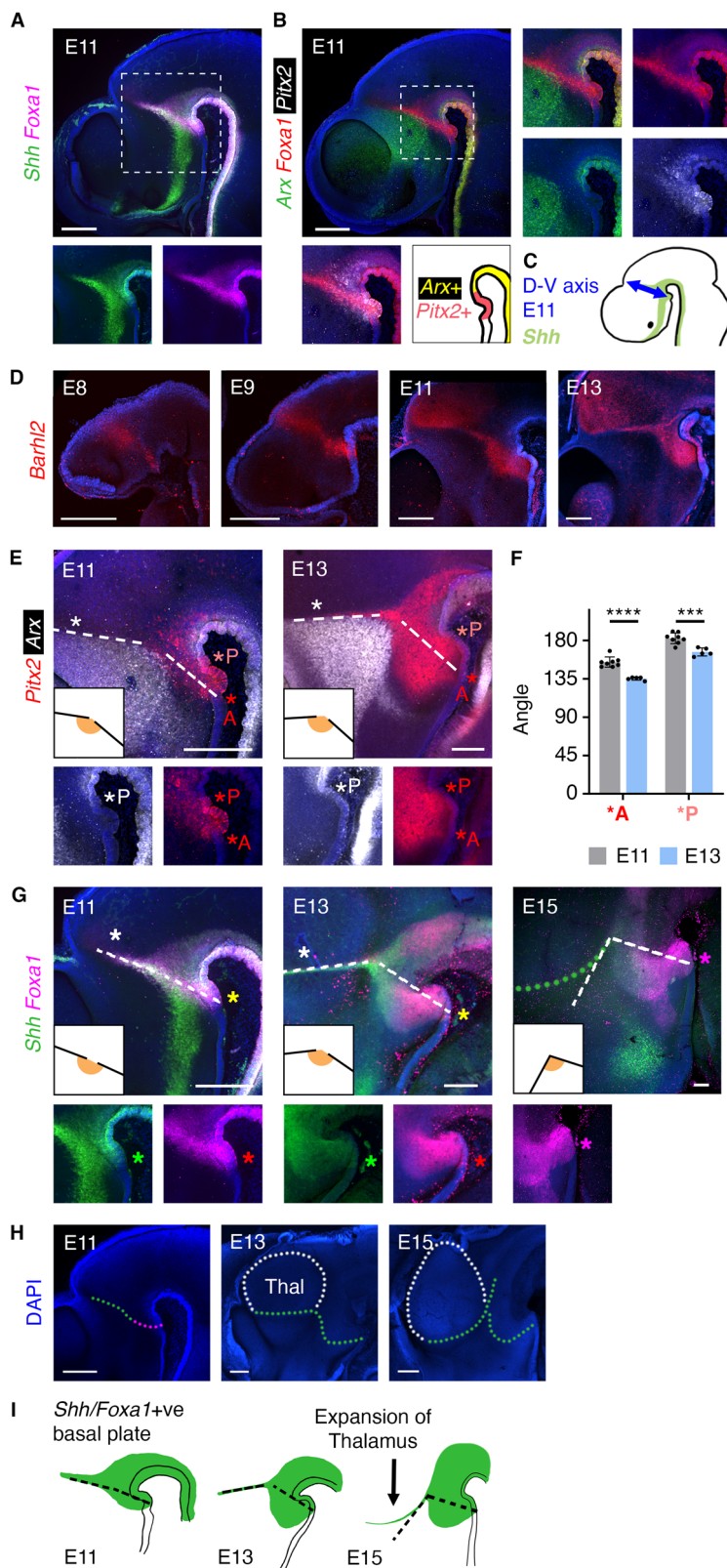

## A possible dorsoventral axis distortion in mouse

We asked whether our findings might translate to mouse forebrain development, analysing selected gene expression patterns in mouse embryos of different stages. Similar to the HH20 chicken, the anterior edge of *Foxa1* forms a nearly straight line, encompassing the *Shh*[+ve] ZLI and continuing to the ventral midline in the E11 mouse (Fig. 4A). *Pitx2* is co-expressed with *Foxa1* ventral to the ZLI and anterior to the *Arx*[+ve]

floor plate (Fig. 4A,B). Thus, our proposed D-V axis (Fig. 3O) translates straightforwardly to the E11 mouse forebrain (Fig. 4C). As the ZLI is intercalated between the prethalamus and thalamus−both diencephalic regions−this implicates the posterior hypothalamus as part of the ventral diencephalon.

In support of this, at E8-E9 a transverse band of *Barhl2* connects ventral posterior hypothalamus with prospective thalamus (Fig. 4D).

**Fig. 4 | Growth patterns may distort the dorsoventral axis in mouse. A, B** HCR-stained E11 mouse brains. Rectangles - positions shown in detail. **B** lower right: *Foxa1*[+ve] ventral midline domains, including *Arx*[+ve] floor plate (*n* = 3). **C** Proposed dorsoventral (D-V) axis, E11 mouse. **D, E, G** HCR-stained E8-E13 mouse brains (*n* = 2 E8, *n* = 3 each, E9-13). Dotted lines in (**E, G**) indicate position of ZLI (white asterisks), and connect base of ZLI to anterior limit of *Pitx2*[+ve] ventral midline (*A in (**E**)) or *Shh/Foxa1*[+ve] ventral midline (coloured asterisks in (**G**)). *P = ventral midline meeting point of *Pitx2*[+ve] posterior limit and *Arx*[+ve] floor plate anterior limit. **F** Angles made between ZLI and lines from base of ZLI to position *A (as per insets in **E**) or position

*P, at E11 and E13. Mean ± standard deviation. *n* = 8 (E11), 5 (E13), ****p < 0.000024, ***p = 0.000463, two-sided t test. Source data are provided as a Source Data file. **H** DAPI-stained E11-15 mouse forebrains, same samples as in (**G**). White dotted lines - outline of thalamus. Dotted lines—anterior edge of *Shh* (green) or *Foxa1* (red) domains in ZLI and posterior hypothalamus. **I** Expansion of the thalamus may displace the *Shh*[+ve] ZLI, causing the angle between the ZLI and posterior hypothalamic *Shh* expression domain to become more acute between E13 and E15. Scale bars − 250 µm.

By E11 an angle develops in the *Barhl2* expression domain, between the ZLI and hypothalamus. To explore whether this may point towards an axis distortion as seen in chicken, we tracked the positions of ventral midline domains with respect to the ZLI, from E10 to E13. Lines were drawn along the ZLI and from the base of the ZLI to the anterior and posterior limits of the *Pitx2*[+ve] ventral midline (Fig. 4E; dotted lines are shown for the anterior markers, *A, and for ZLI position). As in chicken (Fig. 3P), the angles formed between these lines became more acute between E11 and E13 (Fig. 4F). This was also seen when *Shh* and *Foxa1* ventral midline positions were tracked between E11 and E15 (Fig. 4G). Judging by the size of the thalamus, which forms a large bulge at the ventricular surface at E13 and is even more prominent at E15 (Fig. 4H), thalamic growth may contribute to this change by displacing the dorsal ZLI anteriorly. Although it is not possible to resolve this question solely by analysing gene expression patterns, we propose that the D-V axis runs through the ZLI and RM hypothalamus and may become distorted in mice as in chicken.

## The prosomere mamillary domain cannot be identified in the chicken hypothalamus

The D-V axis position we propose in the posterior hypothalamus (Figs. 3Q and 4C) is nearly orthogonal to that in the prosomere model (Figs. 3C and 5A, B). To understand this discrepancy, we conducted a comparative study of prosomere boundaries in chicken and mouse. In the prosomere model, the pretectum, thalamus and prethalamus occupy the dorsal portion of diencephalic prosomeres p1-p3, respectively. The entire hypothalamus is positioned ventral to the telencephalon within two anterior prosomeres (segments), hp2 and hp1 (Fig. 5A)—together comprising the 'secondary prosencephalon'— and the hypothalamus is partitioned along intersecting transverse (interprosomeric; D-V) and longitudinal (A-P) boundaries (Fig. 5B-D). As these boundaries were deduced partly on the basis of mouse gene expression patterns between E11-E19, we first examined defining prosomere regional markers (Fig. 5D) in the E13 mouse brain, focussing on the Periretromammillary (PRM; in hp1), Retro-mammillary (RM, in hp1), Perimammillary (PM, in hp2) and Mam-millary (M, in hp2) hypothalamic regions (see Figure. S13 for comparative hypothalamic terminology, axes, and selected regional marker genes)[6-9,11,23,30].

Confirming previous findings, we readily identified a *Nkx2-1/Sim1/Foxb1*[+ve] (M) domain situated in between *Pitx2/Foxa1*[+ve] (RM) and *Sim1/Otp*[+ve] (PRM/PM) domains, (Fig. 5E-G), including a small *Shh/Foxa1*[+ve] midline region corresponding to the hp2 floor plate (arrowhead, Fig. 5F inset; note the prosomere floor plate is defined as extending beyond *Arx*, up to the anterior limit of *Shh/Foxa1*). *Pitx2* was largely confined to the RM domain, but expression extended into the hp2 floor plate, this analysis revealing that the *Pitx2*[+ve] floor plate underlies both the M and RM domains (white line in Fig. 5G).

However, when we attempted to map prosomere model boundaries onto the HH20 embryonic chicken brain using the same markers, this proved challenging: putative transverse (using *SIM1/NKX2-1/FOXA1*, Fig. 5H, I) and longitudinal (using *OTP/PITX2*, Fig. 5J, K) boundaries appeared to take an identical course, instead of intersecting orthogonally[6,7,23,30], so that *FOXA1* and *OTP* directly abutted (Fig. 5K). Therefore, contrary to published accounts[22,33], a *SIM1/NKX2-*

*1*[+ve], *FOXA1/OTP*[ve] M domain (Fig. 5C,D) could not be identified in the HH20 chicken (Fig. 5L).

Placing prosomere boundaries onto the E7 chick brain was likewise problematic. By E7, *FOXB1* is expressed and a slim *SIM1/FOXB1*[+ve] region could be identified (Fig. 5M), but it overlapped substantially with the RM marker *FOXA1* (compare position of M label in Fig. 5M with RM label in Fig. 5N), and so could not be designated as the prosomere model M domain with certainty. Furthermore, similar to our findings at HH20, putative transverse and longitudinal boundaries (cyan and magenta dotted lines, respectively), intersected obliquely (Fig. 5O, middle panel), the A-P and D-V axes therefore crossing the posterior hypothalamus in similar directions (Fig. 5O, right panel). Notably, the edges of the hypothalamic *Sim1/Otp/Nkx2-1/Pitx2* expression domains are highly curved in mouse, jutting out almost perpendicularly to the ZLI (e.g. Fig. 5E-G). The straighter edges of the equivalent expression domains in chicken more closely resemble those in Xenopus, zebrafish and lamprey[22,34-37], and indeed the E11 mouse (Fig. 4A, B,D). Together this suggests that the morphology of the E13 mouse posterior hypothalamus is derived, the supposedly orthogonal boundaries and axes suggested by the prosomere model attributable to this. We conclude that the prosomere model misinterprets the hypothalamic axes due to an over-reliance on gene expression patterns from a single species at late developmental stages, combined with a flawed growth model arising from a lack of fate mapping information for ventral regions.

## No evidence for a prosomere boundary within the hypothalamus

As our findings challenge both axes and boundaries in and around the M/RM hypothalamus, we expanded our interrogation of the pro-somere model by revisiting the evidence in support of the hp1-hp2 transverse boundary, which purportedly cuts through the hypothala-mus and telencephalon[6,7,23,30]. The position of this boundary was inferred partly from the course of the fornix tract, a structure appearing late in development, and justified by the small M floor plate, which is considered a conceptual requirement for the hp2 segment[6], but the presence of which in chicken we could not confirm. A number of markers distinguishing hp1 from hp2 within hypothalamic long-itudinal zones have been suggested from single colour in situ hybri-disation on mouse brain sections (Fig. S14A,B)[30]. Taking advantage of newer techniques, we applied wholemount multiplex HCR to E13 mouse embryos.

Within the hypothalamus, the hp1-hp2 segmental boundary divides the Paraventricular (Pa) zone into anterior (terminal) and posterior (peduncular) subpopulations (Figs. 5C and S14 A, B). As predicted, the peduncular markers *Lmo4* and *Rgs4* showed a poster-iorly restricted distribution (Fig. S14C,D). The terminal marker *Zic1* was stronger anteriorly, as described, but showed graded expression rather than a distinct posterior limit (Fig. S14C,E). However, the ped-uncular marker *Mfap4* colocalised with the canonical pan-Pa marker *Otp*, rather than being posteriorly restricted (Fig. 5P), and the terminal marker *Fgf15* was expressed throughout the Pa area (Figs. 5P and S14E), as was *Six3* albeit at a very low level (Fig. S14D).

In comparing our results with E13.5 sections from the Allen Brain Atlas[38], we noted that the terminal markers *Fgf15*, *Six3* and *Zic1/5* were expressed most medially (i.e. the ventricular zone), followed by *Otp*,

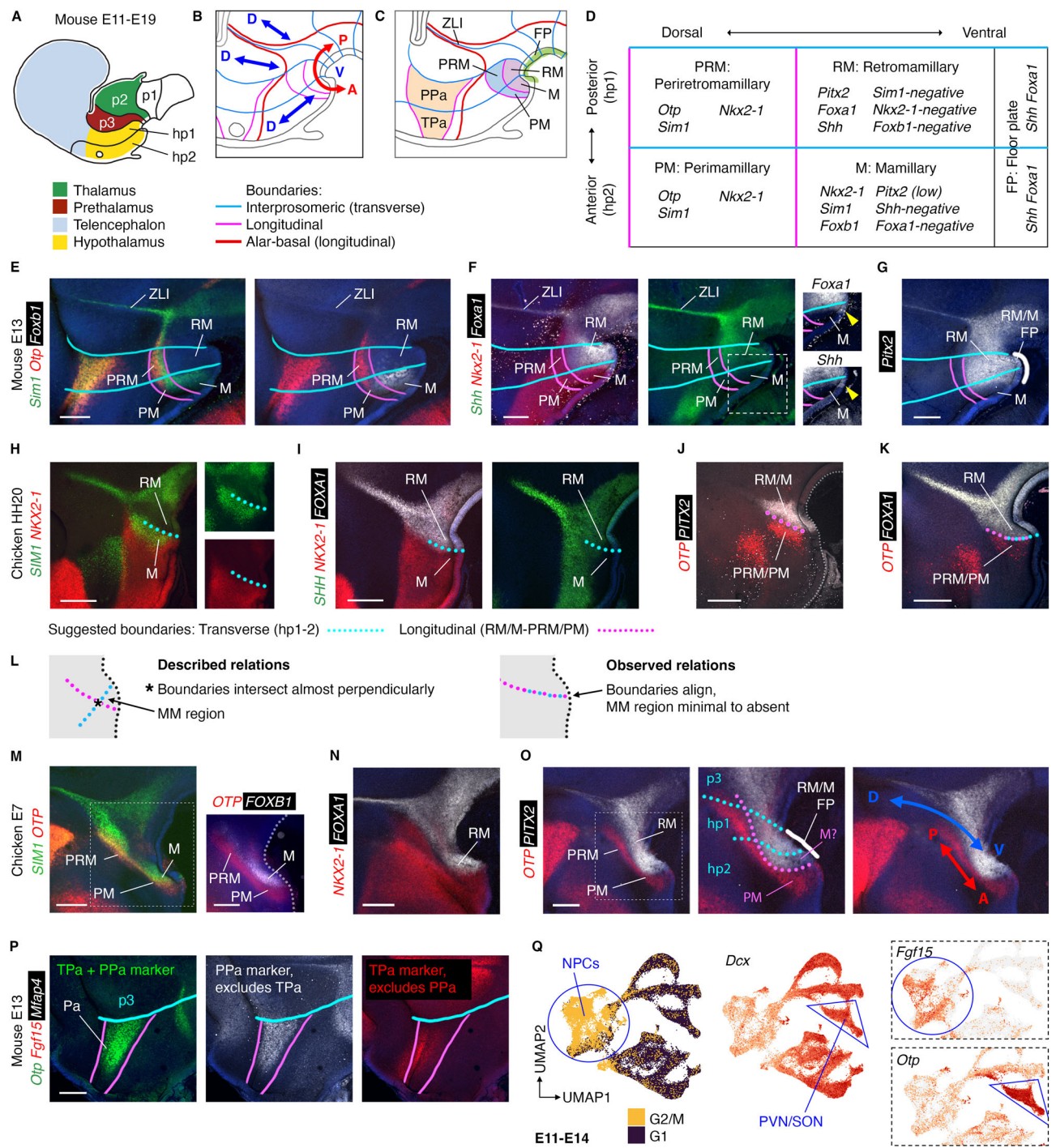

**Fig. 5 | Hypothalamic prosomere boundaries cannot be identified in chicken.**
**A** Mouse prosomeres (2012 model, schematic adapted from[5]) and major forebrain regions. **B** Prosomere boundaries (lines) and axes (arrows) in the hypothalamus and diencephalon. **C** Prosomere hypothalamic regions. **D** Ventral hypothalamic prosomere regions and identifying marker genes. **E-G** HCR-stained E13 mouse forebrains, selected prosomere boundaries overlaid. Dotted rectangle - area shown in **F**. Yellow arrow (**F**) = end of *Shh*⁺ᵛᵉ floor plate. White line (**G**) = extent of *Pitx2*⁺ᵛᵉ floor plate. **H-K** HCR-stained HH20 chicken forebrains, putative prosomere boundaries overlaid. Transverse hp1-2 boundary estimated using *SIM1*, *NKX2-1* (M markers), *FOXA1*, *SHH* (RM markers). Longitudinal boundary estimated using *OTP* (PRM/PM marker), *PITX2*, *FOXA1* (RM markers). **L** Described and observed prosomere boundary relations, HH20 chicken. **M-O** E7 chicken HCRs. Dotted rectangles—areas in **M** (right, different sample) and **O** (middle). Dotted lines—transverse (cyan) and

longitudinal (magenta) boundaries as labelled. Transverse boundaries estimated using *PITX2* (M/RM ventral midline marker), *SIM1*, *FOXB1* (M markers, but see text for discussion), *SIM1*, *OTP* (Paraventricular (Pa) markers). White line—ventral midline *PITX2*. Arrows—forebrain axes inferred from boundary positions. **P** HCR-stained E13 mouse forebrain with selected transverse (cyan, p3-hp1) and longitudinal (magenta) prosomere boundaries. Hp1-hp2 boundary cannot be located based on PPa and TPa markers. **Q** UMAP plots of E11-14 mouse scRNA-Seq dataset[39] showing cell cycle phase (left) and selected genes. Blue circle = neural progenitor cells (NPCs), blue triangle = paraventricular and supraoptic nucleus neurons (PVN/SON). Scale bars − 250 μm. (*n* = minimum of 3 for each HCR probe per species/stage). PPa peduncular Paraventricular, TPa terminal Paraventricular, ZLI zona limitans intrathalamica; for other abbreviations see (D).

*Sim1* and *Mfap4*, with *Lmo4* and *Rgs4* expressed most laterally (Fig. S14F). As proliferative progenitor cells reside at the hypothalamic ventricular surface, the mediolateral distribution of these genes likely reflects a maturity gradient. We sought to confirm this using our recent single cell RNA-sequencing (scRNA-Seq) dataset for pooled E11-E14 mouse hypothalamus[39]. Expression of *Fgf15* was restricted to proliferative neural progenitor cells, while *Otp* was found mainly in differentiating neurons (Fig. 5Q).

To explore whether terminal and peduncular Pa markers may correlate with cell maturity, we extracted and reclustered Pa neural progenitor cells (NPCs) and PV/supraoptic nucleus (PVN/SON) neurons from the scRNA-Seq dataset. Eight clusters resulted (Fig. S15A), of which cluster 6 showed strong enrichment for progenitor/mitotic markers (*Mki67, Top2a, Pcna*; summarised as 'Progenitor score'), the remaining six clusters scoring highly for neuronal markers (*Neurod1, Dcx, Tubb3*; summarised as "Neuronal score', Fig. S15B). The TPa markers *Fgf15* and *Six3* were most highly expressed in progenitor cluster 6, while *Zic1* showed variable expression across the eight clusters (Fig. S15C, D). The PPa markers *Mfap4*, *Rgs4*, and *Lmo4* were also variably expressed, all three genes on balance being lower in cluster 6. Terminal and peduncular 'scores' were calculated based on the TPa and PPa genes displayed in Fig. S15C; progenitor cluster 6 and neuronal cluster 3 scored highest for the terminal profile (Fig. S15B). This suggests that TPa and PPa markers may in part reflect differentiation status rather than segmental identity.

Finally, turning to the Subparaventricular (SPa) zone, we were not able to confirm differential expression of *Vax1* in the TSPa versus PSPa (Figs. S14A and S15E), and *Nkx6-2* was not excluded from the *Six6*⁺ᵛᵉ acroterminal SPa as previously thought (Fig. S15F, see Fig. S15G for the acroterminal concept). While *Meis2* expression did taper anteriorly within the SPa (Fig. S15E), it is unclear why this—or indeed any of the above genes showing graded differential expression—should indicate a segmental boundary, rather than being the result of any other patterning process. Overall, the weak evidence for the hp2-hp1 boundary leads us to question its existence. Along with our axis revisions, these conclusions nullify the key prosomere concept of the secondary prosencephalon and begin to return the hypothalamus to the diencephalon, as per the classical (columnar) view.

## A new model of forebrain developmental organisation

Our results re-orient the position of the D-V axis suggested by the prosomere model, and consequently call into question the A-P axis. The prosomere model takes the anterior limit of ventral midline *Shh/Foxa1* as the anterior limit of the neuraxis. However, while *Shh/Foxa1* expression does indeed terminate ventral to the ZLI in mid-embryogenesis, it extends far more anteriorly at neural tube patterning stages, into and including cells that give rise to the tuberal hypothalamus[13,40]. This suggests that the A-P axis continues into the tuberal hypothalamus. Using our revised axis positions as a starting point, we explored further ways of testing forebrain positional relationships. One approach is to use fate conversion, as cells with similar positional identities often share competence to express a restricted set of factors. We therefore dorsalised the chicken hypothalamus by applying SHH inhibitors (cyclopamine or Sonidegib), with or without follistatin[8], at the neural tube stage and looked for ectopic expression of regional markers.

Dorsalisation treatment at this stage resulted in a small *SHH*⁺ᵛᵉ basal hypothalamus and decreased expression of *NKX2-1* (Fig. S16A,B). The telencephalic marker *FOXG1* was ectopically expressed, but only in part of the anterior-most tuberal hypothalamus (Figs. 6A and S16A-C). Although *FOXG1* is also expressed in the anterior (nasal) retina, no change was detected in the optic stalk marker *PAX2*, suggesting that the converted cells are similar to telencephalon rather than anterior retina (Fig. S16C). Altered cell movements were not responsible for the presence of *FOXG1*⁺ᵛᵉ cells within the hypothalamus, as DiI outcomes

from zone 1/7 injections (in combination with dorsalising treatment) were not altered in position or shape with respect to the optic region (Fig. 6C). This result therefore appears to be a genuine fate conversion of anterior hypothalamus towards a telencephalic identity.

Notably, however, more posteriorly within the tuberal hypothalamus, the prethalamic/PV hypothalamic marker *PAX6* was ectopically expressed (Fig. S16A,C), and the prethalamic marker *ARX* was significantly expanded (Figs. 6A,B, and S16A). Similar results were obtained when HH10 zones 1, 2, 4 and 5 were isolated and cultured in the presence or absence of Sonidegib (Fig. S17). *OLIG2* remained confined to the prethalamus, posterior tuberal and MM hypothalamus (Fig. S16A). Considered alongside our fate map (Fig. 2A, Movie S1), these results suggest that proximity to prospective telencephalon at HH10 predicts competence to express *FOXG1*, and proximity to prospective prethalamus/PV hypothalamus predicts competence to express *PAX6/ARX*. Accordingly, we suggest that the anterior-most tuberal hypothalamus is topologically ventral to the telencephalon, in partial agreement with the prosomere model, but that the remainder of the tuberal region is ventral to the PV hypothalamus and prethalamus, and is part of the diencephalon.

Classically, the PV hypothalamus is included as part of the diencephalon, but the development of the PV hypothalamus remains particularly poorly understood. Our fate mapping revealed a close association between PV progenitors and prethalamus, although growth lines also connect smaller parts of the PV to the telencephalon and basal hypothalamus. Elsewhere in the nervous system, cells with a common identity intermingle, whereas those from different regions segregate. We isolated PV and telencephalic cells (Fig. S18), then disaggregated and reaggregated them in in vitro cultures. PV hypothalamic cells intermingled with prethalamic cells, but segregated from telencephalic cells (Fig. 6D, E combinations A-B, A-C; controls shown in Fig. S19). As prethalamic cells showed poor viability when mixed with telencephalic cells, we could not assess interspersion for this combination (Fig. 6D, E combinations B-C). Similar to the classical view, we therefore tentatively group the PV hypothalamus with the prethalamus as part of the diencephalon, posterior to the telencephalon.

Finally we applied the cell mixing approach to the posterior hypothalamus (Fig. S18), where our D-V axis runs through the ZLI and RM hypothalamus. Dorsally, cells taken from anterior and posterior to the ZLI segregated, albeit only partially (Fig. 6D, E combination B-D. Importantly, basal plate cells taken from anterior and posterior to the RM hypothalamus segregated in culture (Fig. 6D, E combinations E-F and E-G), while those from posterior and anterior or ventral positions within the basal hypothalamus intermingled (Fig. 6D, E combinations G-H, F-G). We therefore suggest that lineage restriction may operate ventral to the ZLI, as shown for an *En1-Dbx1* floor plate boundary in mouse[41].

In conclusion, we forward a 'tripartite hypothalamus' model of forebrain organisation in which the basal hypothalamus sits ventral to three regions—the telencephalon, prethalamus/PV hypothalamus, and the ZLI (Fig. 6F). The *LFNG*⁻ᵛᵉ ZLI and posterior hypothalamus form a topological boundary between the anterior *SIX/FEZ*⁺ᵛᵉ and posterior *IRX*⁺ᵛᵉ forebrain, with the RM hypothalamus part of a ventral boundary region associated with lineage restriction. Informed by high resolution fate mapping, the topological relations of forebrain regions in our model reflect their early spatial origins—and therefore the environment seen by their progenitor cells. Inevitably, emerging and future studies will put this concept to the test.

## Discussion

We have completed the anterior neural tube fate map for HH10 chicken embryos and described the morphogenetic movements shaping the amniote forebrain. Evidence from fate mapping, multiplex HCR analysis, fate conversion experiments and morphology led us to reinterpret the forebrain D-V axis, with direct implications for the

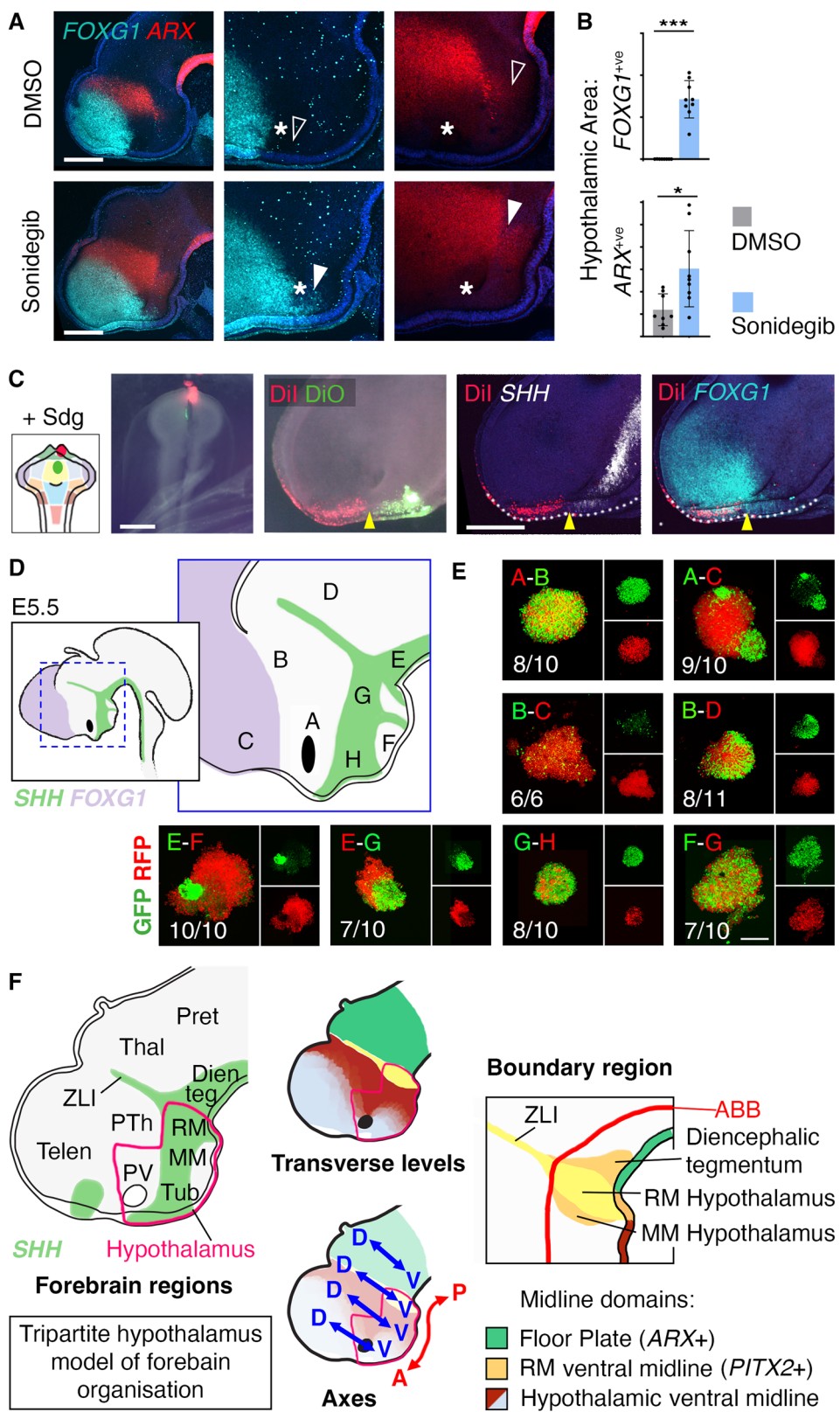

topological position of the hypothalamus, a source of major confusion due to its complexity and relative scientific neglect.

Growth patterns help us to understand the epigenetic history and environmental exposures of cells during a crucial period of specification, relevant for both stem cell differentiation efforts and topological models. Our results reveal early proximity of much of the developing basal hypothalamus to the diencephalic prethalamus and thalamus;

confirm a close association of prospective prethalamus and PV hypothalamus; and suggest that telencephalic and basal hypothalamic lineages are largely segregated. We conducted our fate mapping through patterning stages[8], when cell fate is initially plastic, but becomes progressively restricted according to the positional information encountered. Thus, although it must be stressed that growth lines do not directly delineate forebrain regions, the many similarities

**Fig. 6 | A new model of forebrain developmental organisation. A** Control (DMSO) and dorsalized (Sonidegib) embryos 24 hours after treatment at HH10, showing absent (open arrowheads) and ectopic (filled arrowheads) hypothalamic expression of *FOXG1* and *ARX*. Merged images on the left have been denoised to remove speckles visible in middle panels. **B** *FOXA1*⁺ᵛᵉ and *ARX*⁺ᵛᵉ area within the basal hypothalamus in control- and Sonidegib-treated embryos. Mean ± standard deviation. $n = 8$ (DMSO), 9 (Sonidegib), ***$p < 0.0001$, *$p = 0.0131$, Welch's two-sided *t* test. Source data are provided as a Source Data file. **C** DiI/DiO fate mapping of zones 1/7 in Sonidegib (Sdg)-treated embryos, HH10 and HH10 + 24 hours incubation. Top row—brightfield with fluorescence overlay, bottom row—HCR-stained ($n = 2$). **D** E5.5 chicken forebrain regions targeted in cell mixing experiments. **E** In vitro cultures 48 hours after dissection, disaggregation and mixing of transgenic GFP- and RFP-expressing cells from regions in (**D**). *n* numbers for each condition are indicated, pooled from three independent experiments. **F** Tripartite hypothalamus model of forebrain organisation. Left - HH20 chicken forebrain regions, hypothalamus outlined. Middle—transverse levels and axis orientations; the hypothalamus sits ventral to three regions—the telencephalon, prethalamus/paraventricular hypothalamus, and ZLI. Right—the RM hypothalamus and flanking regions constitute a complex ventral boundary region in line with the ZLI. Scale bars − 250 μm. Dien teg Diencephalic tegmentum, MM Mammillary hypothalamus, Pret Pretectum, PTh Prethalamus, PV Paraventricular hypothalamus, RM Retromamillary hypothalamus, Telen Telencephalon, Thal Thalamus, Tub Tuberal hypothalamus.

in the shapes of DiI growth lines and gene expression domains demonstrate the close coupling of growth and fate.

Notably, shapes similar to our 'growth lines' were seen in an older DiI fate mapping study in the mouse neural tube[42]. Our results are also consistent with cell movements described in the zebrafish ventral midline[43,44], diencephalon[25], and eye[45–48], therefore helping to translate these findings into higher vertebrates. The anterior movement of ventral midline cells that we document here, and in previous work[13,14] likely represents late gastrulation movements, driving the accumulation of tissue into epithelial folds (at HH10) and feeding cells into the basal hypothalamus. The 'wedge'-shaped region encompassing the prethalamus and PV hypothalamus resembles the dorsal portion of the 'D1' neuromere of a previous study[49], and may provide the early morphogenetic basis for the (larger) optic recess region defined in zebrafish[50]. Our results provide important context for understanding eye morphogenesis, a tissue that is very challenging to fate map owing to the rapid and extreme tissue deformations involved.

We propose that the anterior tuberal hypothalamus is situated ventral to the telencephalon, and that the posterior tuberal and mammillary hypothalamus are ventral to the PV hypothalamus and prethalamus (Fig. 5K). We do not propose a 'hard' (lineage restricted) boundary between anterior tuberal/telencephalon and more posterior regions, and view these relationships as topological rather than segmental. Our model builds on accumulating evidence, as the close relationship of much of the hypothalamus to prethalamus is well supported by both genetic and biochemical analysis[8–12,25].

We place the RM hypothalamus ventral to the ZLI and hence posterior to the prethalamus. We note that *BARHL2* (in chicken) traces the edge of the ZLI and RM hypothalamus and may therefore delineate a topological boundary, separating *IRX*⁺ᵛᵉ from *FEZF*⁺ᵛᵉ regions. We speculate that the RM hypothalamus and flanking *WNT8B*⁺ᵛᵉ regions represent an extensive ventral 'boundary region' (Fig. 5K), where the juxtaposition of anterior (*FEZF*⁺ᵛᵉ) and posterior (*IRX*⁺ᵛᵉ) forebrain - plus underlying tissues such as notochord and prechordal mesoderm - provided an evolutionary substrate for the generation of high signalling and cell type complexity. Differential expression of Notch and Ephrin pathway components, secretion of signalling ligands and cell sorting experiments link this area to boundary functions: boundaries frequently function as signalling centres during development, and signalling may contribute to their ongoing maintenance[51]. Cells originating in the RM region contribute widely to the posterior hypothalamus and beyond, with *PITX2/LMX1B*⁺ᵛᵉ progenitors in mouse contributing to the supramammillary, subthalamic and ventral premammillary nuclei[9,29,41,52,53]. We therefore accept the subthalamic nucleus as part of the hypothalamus, a view that has gained considerable traction recently[6,54].

The RM ventral midline is distinguished from floor plate by *PITX2* and the absence of *ARX* (Fig. 5K), and is likely conserved to mouse[41] and human[3]. The whole hypothalamic midline lacks a floor plate at mid-embryogenesis but arises from *Foxa1*⁺ᵛᵉ *Shh*⁺ᵛᵉ floor plate-like (HypFP) progenitors[13,55], its uniqueness accounted for by the distinct

evolutionarily origin of the anterior forebrain[56–59], and mechanistically by the distinct patterning influence of the prechordal mesoderm[60–63]. We do not define a specific point in the anterior tuberal hypothalamus where the ventral midline ends anteriorly, but note that in principle it could extend far enough to abut the ABB.

The defining feature of our model is that the ZLI and ventral boundary region cut through the diencephalon, dividing the forebrain into anterior and posterior parts. The hypothalamus is therefore placed ventral to the telencephalon *and* to two parts of the diencephalon (prethalamus and ZLI), hence 'tripartite'. Therefore, the majority of the hypothalamus is considered ventral diencephalon, similar to classical columnar views. However, we have adopted the curved ABB from the prosomere model, and link the anterior tuberal hypothalamus and telencephalon. (We also note that earlier versions of the prosomere model placed axes more similarly to ours[64]). We now use the term retromammillary for the progenitor zone we previously referred to as supramammillary[8], agreeing that this is posterior to the mammillary region, and wishing to distinguish it from its major derivative, the supramammillary nucleus. Therefore, our model incorporates elements from both columnar and prosomere models. Fundamentally, however, we do not consider the anterior forebrain (prosomere model p3, hp1, hp2) to be segmented as suggested by the prosomere model. And whereas the prosomere model claims a segment boundary between the prethalamus and PV hypothalamus, we group these regions, reflecting the intimate connection of their progenitors and relative separation from telencephalon by the eye field. Indeed, we further speculate that the eye itself has a 'split' identity, the anterior (nasal) side relating to the telencephalon, and the posterior (temporal) side relating to the prethalamus and hypothalamus—consistent with the expansion of both telencephalon and prethalamus into the eye field of *Lhx2* and *Rax* mutants[12,65].

Critically, our model is informed by evidence from early developmental stages, minimising the confounding effects of neuronal migration. All major progenitor zones are established in the chicken hypothalamus by the experimental endpoints used[8], enabling the inference of adult nuclei origins by comparison to longer term fate mapping studies and developmental single cell sequencing datasets, including to some extent cross-species data. Continued rapid progress in understanding the ventral forebrain in particular should confirm these inferences, directly or indirectly. Our results offer new insights into forebrain morphogenesis and organisation and should help to inform efforts to direct pluripotent stem cell differentiation towards specific forebrain identities.

## Methods

### Embryo sourcing and incubation

Fertilised Bovan Brown eggs (Henry Stewart & Co., Norfolk, UK), and transgenic *Chameleon* (cytbow)[15], Cytoplasmic GFP[66], and Flamingo (tdTomato)[15] eggs (The Roslin Institute, Edinburgh, UK) were incubated in a humidified incubator at 37 °C until the desired stage, according to[67,68]. Time-mated pregnant wildtype C57Bl/6 mice were sourced from Envigo RMS (UK) Limited, Shaws Farm, Bicester.

## Immunolabelling

Embryos were fixed in 4% PFA for 2 hours at 4 °C and washed in PBS. Samples were incubated in block solution (1% TritonX, 10% heat-inactivated goat serum (HINGS) in PBS) overnight, then primary antibody solution (Rabbit anti-laminin α (LAMA), Sigma L9393 at 1:1000 in block solution) overnight. Embryos were washed twice in PBS at room temperature, and transferred to 10% HINGS in PBS and left overnight. Alexa 488-conjugated anti-rabbit IgG secondary antibody (1:500, Jackson ImmunoResearch) and phalloidin-TRITC (1:100, 1% TritonX, 1% HINGS in PBS) were applied at 4 °C overnight, washed the following day in PBS at room temperature, then washed overnight in PBS with 0.1% TritonX. All steps were performed at 4 °C except where stated.

## DiI/DiO injections

Eggs were windowed and Coomassie Blue (0.5 µl/ml in L-15 - Fisher Scientific, Cat No 11580396) was injected under the embryo to aid visualisation. The forebrain inner neuroectoderm was accessed via a dorsal midsagittal incision. CellTracker™ CM-DiI Dye (Invitrogen, Cat No. C7000) in ethanol (50 µg per 30 µl), or Vybrant™ DiO cell-labeling solution (Invitrogen, Cat No. V22886) was loaded into a fine glass needle and injected into embryos by hand using a Parker Picospritzer II (10-20 msec pulses, 15 psi). Dye location was noted and imaged using a Leica MZ16F stereomicroscope. Embryos were incubated for 48 h or as stated, fixed in 4% buffered paraformaldehyde, hemisected and reimaged prior to processing for HCR. Some DiI/DiO signal is lost during the HCR procedure, therefore dye and gene expression patterns are shown separately for some examples.

## HCR analysis

Embryos were fixed overnight in 4% paraformaldehyde at 4 °C and stored in methanol for a minimum of one night before rehydrating in PBS. HCR v3.0[69] was performed on intact embryos according to the manufacturer's RNA-FISH protocol for whole-mount chicken and mouse embryos (as shown on Molecular Instruments website), using reagents obtained from Molecular Instruments, Inc. (Los Angeles, CA, USA). HCR probes were designed by the manufacturer based on the following accession numbers or the manufacturer's database ('infinite catalogue'): Chick: *ARX* (XM_025146483.1), *BARHL2* (XM_015290665.4), *EMX2* (XM_025152058.1 and XM_025152057.), *EPHA7* (NM_205083.1), *FEZF2* (XM_414411.5), *FOXA1* (XM_004941922.3), *FOXA2* (NM_204770.1), *FOXD1* (NM_205192.3), *FOXG1* (NM_205193.1), *IRX3* (XM_015292372.4), *LFNG* (NM_204948.1), *LHX6* (XM_015279838.2), *LMX1B* (NM_205358.1), *NKX2-1* (NM_204616.1), *NKX2-2* (XM_015283379.2). *OLIG2* (NM_001031526.1 with extra sequence from NC_006088.5 | :106522977-106525323 #256 chromosome 1, GRCg6a), *PAX6* (NM_205066.1), *PITX2* (NM_205010.1 with additional 5' exons from XM_025149516.1/ XM_025149515.1), *SHH* (NM_204821.1), *SIM1* (XM_004940357.3), *SIX6* (NM_001389365.1), *WNT8B* (XM_025151998.1).

Mouse: *Arx* (NM_001305940.1), *Barhl2* (NM_001005477.1), *Fgf15* (Infinite catalogue), *Foxa1* (NM_008259.4), *Foxb1* (NM_022378.3), *Lmo4* (Infinite catalogue), *Meis2* (Infinite catalogue), *Mfap4* (Infinite catalogue), *Nkx2.1* (NM_009385.4), *Nkx6.2* (NM_183248.4), *Nr5a1* (Infinite catalogue), *Otp* (NM_011021.5), *Pitx2* (NM_001287048.1), *Rgs4* (Infinite catalogue), *Six3* (Infinite catalogue), *Shh* (NM_009170), *Sim1* (NM_011376.3), *Vax1* (Infinite catalogue), *Zic1* (Infinite catalogue).

Proteinase K treatment (10 µg/ml) was performed for 2-30 min and embryos were re-fixed for 20 minutes, washed in PBST and transferred to 5x SSCT on ice. Embryos were preincubated in hybridisation buffer for 30 mins, then hybridised overnight in 10 nM (1:100) probe solution in hybridisation buffer, both steps at 37 °C. Samples were washed 4 × 15 minutes in wash buffer at 37 °C, then 2 × 5 minutes in 5x SSCT at room temperature. Embryos were equilibrated in amplification buffer for 5 minutes before adding the hairpins solution. Even and odd hairpins were melted (90 secs, 96 °C) and cooled

(30 minutes, room temperature) separately before mixing in amplification buffer (1:50). Samples were incubated in the dark overnight and washed in 5x SSCT (2 ×5 mins, 3 ×30 mins, 1 ×5 mins). Embryos were hemisected or the neuroectoderm isolated (see below) as necessary. Tissue was counterstained with DAPI prior to imaging (Cell Signaling Technology, Cat no. 4083, 1:1000).

## Quantification of DiI/DiO results

Seventeen DiI/DiO outcome areas (Fig. 1H), and the growth line shapes associated with each (Fig. 1G), were developed based on extensive analysis of >300 DiI/DiO patterns. Areas were named according to the major region in which they were located (T1-T2 - telencephalon, D1-D7−diencephalon excluding hypothalamus, H1-H5−hypothalamus and Hm1-Hm3−hypothalamic midline), but as these were based on DiI/DiO patterns and not gene expression, in some cases the areas overlap more than one major region.

Quantification was performed only on samples injected at HH10. Positional outcomes were scored by assigning each result to one or more of the seventeen outcome areas. For each of the twelve injection zones, and the meeting points of adjacent zones, the percentage of samples showing labelling in each area was calculated. Results were displayed as thumbnail images where all outcome areas with at least one instance were coloured according to frequency, i.e. white means no examples were seen. Concurrently with assigning positional outcomes, it was judged whether the shape of each resulting DiI/DiO pattern (growth line) conformed to the description for each area. In some cases the location of labelling was clear but the shape of the growth line was not; these cases were pooled with those examples where the described shape was not seen.

## Cre recombination and neuroectoderm isolation

Chameleon (Cytbow) eggs were incubated until HH9, windowed, staged, and screened for the Cytbow transgene, using a handheld 365 nm black-light torch. 0.5-1.5 µl TAT-Cre recombinase (1500U) (Merck Life Science UK, SCR508) diluted 1:20 to 1:200 in L-15 was injected into the anterior neural tube using a pulled capillary needle. Eggs were sealed and incubated for a further 24-48 hours before fixing and staining for *NKX2-2/WNT8B* by HCR. Following HCR, forebrains were hemisected and the neuroectoderm isolated by microdissection aided by treatment with Dispase (1 mg/ml) for approximately 10 minutes to help separate tissue layers. Highly curved regions were nicked with a scalpel to enable flat mounting on a standard microscope slide.

## Electroporations

A Microelectrode holder (World precision Instruments, MEH6RFW) with a 0.25 mm silver wire was used as the cathode. A capillary needle containing 1-2 ug/ul pCAGGS-RFP with 0.1% Fast Green dye was threaded over the silver wire, and a mouth pipette attached to the side inlet valve to control the internal pressure. The anode, made from a tungsten needle, was positioned under the target cells and 2 × 3 pulses of 40 V were applied using a TSS20 Ovodyne electroporator (Intracel), whilst simultaneously increasing the pressure in the needle to release the DNA mixture. Eggs were then re-sealed and incubated for a further 24-48 hours.

## Imaging and image processing

Images were taken on a Zeiss Apotome 2 (10x objective) with ZEN2 Pro, blue edition software (Zeiss), Leica MZ16F using LAS X 1.1.0.12420 software, Zeiss Lightsheet (5x, 10x objectives) with ZEN software (Zeiss), and Nikon W1 Spinning Disk confocal (4x, 10x, 20x objectives) with Nikon NIS-Elements AR version 5.42.03 (build 1812) software. For Apotome, Lightsheet and Spinning Disc imaging, embryos were mounted in 1% agarose, and z-stacks are presented as maximum intensity projections. Image processing was performed using Fiji (ImageJ, version 2.7.0/1.53t) and Adobe Photoshop 2022.

Hemisected embryos are presented as the right side of the embryo, for ease of comparison. Gamma levels have been altered for clarity.

## 4-dimensional modelling of forebrain morphogenesis

Neuroectoderms were isolated (see above) from HH10, HH11, HH14 and HH20 chicken embryos, stained with DAPI, and imaged on a Zeiss Lightsheet confocal (Fig. S6, step 1). Image stacks were converted to binary, denoised, and exported as a wavefront (.obj file) using FIJI software and the 3D viewer plugin (Fig. S6, step 2). The.obj was imported into the open source animation software Blender (blender.org) and the smooth tool (in sculpting mode) was used to remove the artefactual contour lines. The objects were remeshed to reduce file size, and these meshes were used as morphological templates to build the model around (Fig. S6, step 3).

A simple mesh was built around the HH10 template by first adding a single plane and enabling snapping to the surface of the template (using Face Project, with 'Project individual elements' enabled, as well as the Shrinkwrap modifier; Backface culling was enabled, as was 'in front' under Object Properties/Viewport Display). Geometry was added using the modelling tools, until the entire right hand side of the neuroectoderm was represented by a simple quad mesh. The mirror modifier was used to view the entire neuroectoderm to assist progress, but never actually applied to the mesh. Different faces of the mesh were given different colours, to assist subsequent stages by making them more easily identifiable. The simple quad mesh was then duplicated (ensuring the new object was not linked to the original) and snapped to the surface of the HH20 template mesh. The vertices were moved using the tweak tool, so that their positions at HH10 and HH20 roughly corresponded, according to the DiI/DiO results from the present study and the fate maps of Garcia-Lopez et al.[16] and Pombero et al.[17]. In a similar manner, the meshes were copied and snapped to the HH11 and HH14 template meshes, and the vertices positioned to intermediate points (between the HH10 and HH20 meshes) (Fig. S6, step 4). The meshes were periodically subdivided by applying the subdivision modifier (simple setting, Fig. S6, step 5) and adjusting further, until the level of detail and precision, was deemed sufficient. All such changes to the mesh geometry were applied identically to all four meshes in order to enable joining later. Progress was monitored along the way by joining the meshes (usually only the HH10 and HH20 meshes) as shape keys ('Join as shapes' option); selected faces were then coloured according to HH10 position or HH20 region, and the outcome at the opposite timepoint was viewed to assess the fate map or 'digital dye' growth lines.

Once the vertex positioning was complete, the four meshes were duplicated to the insides of the template meshes and adjusted for optimal fit. Inner and outer meshes were joined and subdivided further, so that they closely followed the surfaces of the template meshes (Fig. S6, step 6). The four meshes were then joined as shape keys to enable warping between the four stages (Fig. S6, step 7). Fate map and 'digital dye' colours were applied to the surface by creating a UV map and using the texture painting options (Fig. S6, step 8). A procedural material was assigned to generate a surface texture. Forebrain animations were created and exported from Blender, and annotated movies were compiled using Adobe Premier Pro 2024 software. Note that the modelling approach does not allow for any 'cell mixing' between adjacent parts of the mesh, hence 'digital growth lines' are slightly smaller than in vivo.

## Morphometric analysis

To measure chicken forebrain dimensions (Fig. S10), HH10-HH19 forebrains were stained by HCR for *FOXG1*, *OLIG2* and *WNT8B* and hemisected and imaged on a Nikon W1 Spinning Disc confocal. Morphometric measurements were performed on maximum projections of z-stacks using FIJI (ImageJ, version 2.7.0/1.53t). Distance measurements were obtained using the line tool and 'measure' function. First, line A (Fig. S10B) was drawn between two trackable points X and Y: respectively, the midline at the ventral limit of $FOXG1^{+ve}$, and a corner of the *OLIG2* expression domain (Fig. S10B-C); our DiI results demonstrate that these points correspond at HH10 and HH20 (X - zone 1/7 boundary, Y - zone 6, Fig. 1D). A perpendicular line was then positioned to intersect both line A and the posterior margin of the optic vesicle/stalk, which was clearly visible in hemisections as an edge (brightness/contrast levels were adjusted as needed). Line B was drawn between the posterior limit of A and the point of intersection of A and the perpendicular line. Results were exported to Excel using the Read_And_Write_Excel Plugin, the B:A ratio was calculated, and a logarithmic trend line was added.

To track mouse dorsoventral axis distortion (Fig. 4E-G), lines were drawn on HCR-stained mouse E11/E13 forebrains along the ZLI, and from the base of the ZLI to the ventral midline: point *A marked the anterior limit of *Pitx2* ventral midline expression, and point *B marked the *Pitx2* posterior limit and the *Arx* anterior limit. Angles between the two lines were measured using the angle tool in Adobe Photoshop. Two-sided t tests were run using Graphpad Prism 10.6.1.

## scRNA-Seq analysis of mouse E11-14 PVN lineage

We analysed single-cell RNA-Seq (scRNA-Seq) resources generated previously in the lab, a whole-hypothalamus E11-E14 10x Genomics dataset (GEO accession GSE284492), to subset paraventricular/ supraoptic neurons and their local progenitors ("PVN + NPC"). All analyses were performed as previously described[39].

Cells expressing *Otp* and/or *Sim1* were retained, but NPCs lacking these markers were also included to represent earlier stages of the lineage. Contaminating ventral/posterior populations were removed by excluding any cell with >0 UMIs for *Foxa1*, *Foxa2*, *Pitx2*, or *Shh*. Additional marker-based filtering removed cells expressing *Nr5a1*, *Foxb1*, *Tbr1*, *Agrp*, or *Pomc*. The remaining cells were reclustered de novo and visualised using UMAP. Gene-set module scores were computed using Seurat's *AddModuleScore()* function[70]. The anterior/terminal (hp2) module included *Fgf15, Six3, Zic1*, and *Zic5*, while the posterior/peduncular (hp1) module included *Mfap4, Lmo4*, and *Rgs4*. To capture proliferative and differentiating states, a cycling progenitor score was calculated using *Mki67, Top2a*, and *Pcna*, and an early neuronal differentiation score was computed using *Neurod1, Dcx*, and *Tubb3*. For each cell, genes of interest were overlaid on UMAPs or summarised in dot plots.

## Dorsalisation of chicken embryos

Embryos were windowed at HH10-11 and the vitelline membrane torn above the head. Dorsalisation was induced by treatment with the SHH antagonist Sonidegib (Cat No.S2151, Selleckchem). 2 mM in DMSO was diluted 1:4 in L15 to 0.5 mM, or DMSO only diluted 1:4 in L15, and 2ul pipetted onto embryo and the egg re-sealed. Alternatively, Cyclopamine (1 μg/μl; Sigma - C4116, dissolved in PBS with carrier 45% 2-hydroxypropyl-β-cyclodextrin, Sigma - H5784), with or without the TGF-β inhibitor Follistatin, recently shown to repress ventral identity[8] (15 μg/μl Recombinant Mouse Follistatin 288; 769-FS, Bio-Techne). 5 μl of cyclopamine only or 5 μl of Cyclopamine/FST (1:1) was pipetted on top of the embryo and the egg was re-sealed. Embryos were incubated at 37 °C and were dissected and fixed 24-48 hours later and processed for analysis by HCR.

Quantification: Maximum projections of spinning disk images taken at the same settings were opened in Fiji and the selection tool used to manually select the hypothalamus, the resulting shape saved in the ROI manager. *FOXG1* HCR images were de-speckled using the Fiji Remove outliers tool before applying an equal threshold to all images. The hypothalamus ROI was reapplied and the area of positive expression within measured. *FOXG1* expression has been de-speckled

Article

using the photoshop 'dust and scratches filter' in Fig. 6A panel 1, but the same image shown at higher magnification in panel 2 has been left unaltered. Two-sided Welch's $t$ tests were run as the variances differed between groups ($p < 0.00001$ FOXG1, $p = 0.0333$ ARX), using Graphpad Prism 10.6.1.

## Cell mixing analysis

Transgenic Cytoplasmic GFP and Flamingo (tdTomato) chicks were dissected at E5 or E6 into L-15 on ice. Neuroectoderms were isolated following Dispase (Sigma-Aldrich, Cat No. 4942086001) treatment and forebrain regions microdissected. A set of explants from individual forebrain regions were immediately fixed, and processed for HCR in situ to confirm accuracy of dissection. A second set of explants from each region (unfixed) were incubated in Papain (2 mg/ml) for 10 mins at 37 °C and dissociated into single cells by trituration using fire polished glass pipette with different capillary openings (large and small). Following dissociation, the enzyme was inactivated with 5 mls of L-15 and cells gently centrifuged at 600 RPM for 5 mins at 4 °C. Cells were resuspended in 1 ml of explant medium[71] and counted using a haemocytometer. Hanging drops were formed from 15ul cell suspension containing 20,000 cells/drop, and incubated at 37 °C for 24-48hrs to form pellets. Cell pellets were fixed, cryosectioned and imaged without further staining.

## Explant culture

HH10 neuroectoderms were isolated following dispase treatment (Cat no 4942086001, Roche, 1 mg/ml in L15 medium at room temperature, 5-15 minutes)[72]. Explant encompassing zones 1-2-4-5 was carefully sub-dissected and mounted on to matrigel (11523550, Fisher Scientific UK) beds and cultured for 24hrs at 37 °C[71]. Explants were treated with DMSO control or Sonidegib (200 μM). Explants were then fixed and processed for analysis by HCR.

## Ethics

All work was performed at our licensed establishment (UK Home Office under the Animals (Scientific Procedures) Act 1986) and was approved by the University of Sheffield Animal Welfare and Ethical Review Body (AWERB). Named Animal Care and Welfare Officers (NACWOs) had oversight of all incubated eggs. Pregnant female mice were killed by a Schedule 1 method.

## Reporting summary

Further information on research design is available in the Nature Portfolio Reporting Summary linked to this article.

## Data availability

The authors declare that the minimum dataset that is necessary to interpret, verify and extend the research in this article is included within the manuscript and its supplementary information files. All scRNA-Seq data used in this study were previously generated and published in Kim et al. (2025), and deposited in the NCBI Gene Expression Omnibus (GEO) under accession number GSE284492. Source data are provided with this paper.

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

## Acknowledgements

The authors thank Alex Fletcher for comments on the manuscript, and Nick Van Hateren and Darren Robinson for help with imaging and Blender. Imaging work was performed at the Wolfson Light Microscopy Facility using the Zeiss Light sheet (BBSRC BB/MO12522/1) and Nikon spinning disc (BBSRC2020 Alert B/V019368/1) microscopes. Funding was provided by Wellcome Trust grant 212247/Z/18/Z (to MP), Wellcome Trust grant 303188/Z/23/Z (to M.P.), and NIH grant R01MH126676 (to S.B.). Open Access funding provided by the Wellcome Trust. Deposited in PMC for immediate release.

## Author contributions

Conceptualisation: E.M., S.B., M.P., E.P. Investigation: E.M., K.C., C.F., D.W.K., M.P., E.P. Visualisation: E.M., K.C., C.F., D.W.K., E.P. Formal analysis: E.M., D.W.K., M.P., E.P. Validation: E.M., M.P., E.P., Data curation: E.M., D.W.K., M.P., EP., Project administration: M.P., Writing—original draft: EP, Writing—review: EM, MP, EP, Writing—editing: E.M., K.C., C.F., D.W.K., S.B., M.P., E.P.

## Competing interests

The authors declare no competing interests.
