## [Transparent Peer Review file · Nature Communications]

Resolving forebrain developmental organisation by analysis of differential growth patterns

Corresponding Author: Professor Marysia Placzek

Version 0:

Reviewer comments:

Reviewer #1

(Remarks to the Author)

This is a very useful resource for the developmental neuroscience research community. The authors present a set of nice lineage tracing experiments in the chick forebrain. Based on those, 3D computational models (fantastic movies describing likely 3D temporal progression of domains) and restricted set of HCR mRNA detection, the authors refute the current prosomeric model of forebrain organisation, proposing a modified model. Although the work is very carefully done and of general interest for the field, we have difficulties with the lack of focus of the narrative and some lack of evidence for the model proposed (while agreeing on finding the prosomeric model problematic for the anterior and ventral forebrain). Overall, the work as it is presented currently does not support strongly enough the conclusions made by the authors.

Specific points to be addressed:

- The most precious part of the data is the collection of lineage tracing results. Although the stated n for each of the injected forebrain areas described on Figure 1 is high, there is no quantification of the clone size, shape and position at HH20 across embryos injected for each area at HH10. This quantification is essential to assess variability across injections. Figure S2 suggests a fair amount of variability and same embryos are used for two distinct data points, using different left and right positions of injection (eg. Fig. S2 E & F, H & I, J & K).
- The last frame of Movie S2 should be used in an improved Figure 2 to visually match the findings with the model of clonal shapes (if the above quantification of clone size, shape and final position confirms the current conclusion per position on the HH10 map).
- The dynamic model for the yellow area in Movie S4 (not equivalent to the yellow area of Figure 1) is key to the author's model but does not have enough lineage tracing evidence for its proposed origin. Specific lineage tracing of this area (with enough n) is needed.
- The attempt to refute the prosomeric model is not well articulated, offering partial evidence for the proposed alternative. The most important data would be extensive tracing in area 1, 2, 5 and 7. Fig S4D is key and yet is not showing the position of the injected cells at HH10.
- A substantial portion of the argument for the proposed 'compartmental' origins of the hypothalamic areas is based on HCR analyses. However, dynamics in gene expression can't be used as rigorous support for clonal dynamics. This is a big limitation of the manuscript. Higher resolution lineage tracing is needed to back the HCR conclusion up.
- The cell mixing experiment (Fig 4J), where cells separate based on their positional identity (anterior hypothalamic vs posterior thalamic), is key to the proposed model and is missing key controls and ambition. It would need i) mixing the posterior hypothalamic cells with their own (A with A in 4J, matching the P with P) and with cells from the proposed anterior hypothalamic compartment, to see if mixing results match their proposed model.
- The quality of the Dil/DiO images is very low. And the Cre/tdTomato images are also very hard to interpret because of overabundance of positive cells. A better approach could be to inject fluorescently labelled viruses in regions of interest and see lines of cells that way.

- The point about dorsalization of ventral diencephalon region with cyclopamine is weak, as the markers they use to indicate telencephalon (Foxg1) and prethalamic/PV hypothalamic (Pax6) are also expressed in the eye. Given that this treatment is known to lead to cyclopic phenotypes, the markers may be present in 'trapped' retinal cells, instead of proving a fate conversion to TC and pre-thalamus. Much more needs to be done to conclude anything from this experiment.
- The comparison between chick and mouse is potentially powerful but only basing it on HCR of a few markers introduces too many possible caveats and shouldn't be considered here unless adding clonal evidence.

Smaller issues:

- They refer to figure 5K in the discussion. That panel doesn't exist in figure 5.
- Figure S5 is not useful as the Cre clones can't really be informative if not quantified carefully (size and shape) across many embryos.
- Some of the HCRs are hard to see especially when they've used grey colour on top of the faint blue background.
- The most important focus of the manuscript is the organisation of the hypothalamus. Yet, the results are written in a very complicated fashion, distracting the reader from the most important conclusions. We would advise to reshape the text to facilitate understanding and clarity of thoughts.

Reviewer #2

(Remarks to the Author)

This manuscript describes a comprehensive and detailed fate-mapping of the chicken forebrain at HH10 using fluorescent dye labeling along with spatial expression pattern analysis of multiple marker genes.

The following new knowledge has been generated from this study:

- 1) Generation of detailed fate map of the chick forebrain at HH10.
- 2) Generation of a 4D model of forebrain growth.
- 4) Establishing that eye morphogenesis plays a key role in separation of the telencephalon from the prethalamus and the hypothalamus.
- 3) Proposal of a "tripartite" model for the hypothalamus.
- 4) Identification of a major forebrain boundary that cuts through the MM hypothalamus and the ZLI.

This study does not contain any functional characterization that may shed light on the genes and mechanisms involved in the morphogenesis of the forebrain, particularly the hypothalamus. However, it has brought significant clarity to this phenomenon with respect to the growth movements and the changes in the orientation and positioning of the AP and DV axes. In fact this study has challenged some aspects of the existing prosomere model and proposed a new "tripartite" model for the formation of the hypothalamus. Thus significant new knowledge has been added to the field.

The experiments that have been carried out largely support the conclusions made and the data has been analyzed in depth and interpreted correctly.

The methodology used meets the expected standards in the field and sufficient details have been provided in the supplementary materials for reproduction.

Hence this manuscript is suitable for publication without the requirement of any major changes.

Reviewer #3

(Remarks to the Author)

The authors of "Resolving forebrain development organization by analysis of differential growth patterns" proposed a common anterior compartment of the forebrain with telencephalon, hypothalamus and prethalamus. In addition, the authors claim a new model of forebrain development with a tripartite hypothalamus. This is a detailed study that provides extensive data that aims to support their hypothesis. However, after a detailed analysis of the manuscript, some key points need to be resolved by the authors, including the need to better clarify whether they are proposing a new model or suggesting some changes to the prosomeric model, but also other conceptual and interpretive points (see below).

The main novelty of this work is the improvement of fate map data at stages that have not been sufficiently explored or resolved in the chicken. Fate map information is an essential tool to resolve questions about the specific origin of cell populations and addresses questions such as those raised here. These fate map data compared with molecular data provide the authors with the possibility of reaching some conclusions that are interesting, but not definitive and that in some points require a deep understanding and exploration due to the complexity of the adult brain that seems not to be sufficient in the proposed analysis at these early stages (see below for specific details). One of the controversial aspects supported by the authors is their claimed ZLI distortion. Although the work does not definitively demonstrate its proposal, it is an interesting

alternative that should be taken into account to definitively resolve this problem in the future.

General comments:

I need to highlight three main aspects that the authors need to improve in the manuscript, one related to the model, another related to the conceptual assumption of the hypothalamus and third one the validity of the selected strategy to resolve the fate map and combining it with the molecular map.

The first point is related to the model. There are two main and somewhat antagonistic models to understand how, starting from the stages of the neural plate and going through the different stages of the neural tube, the brain is parceled out and the main derivatives of each region have emerged. These models are the columnar model and the prosomeric model. Both models are different because, for example, they assume different axial dimensions that determine different dorsoventral relationships along the anteroposterior axis, but in addition, the prosomeric model establishes an anteroposterior segmentation in terms of neuromeres. These differences concern, for example, the relationship of the roof and floor plates to the components of the neural tube (for details of both models, see, for example, in Puelles 1995; A segmental morphological paradigm for understanding vertebrate forebrains). If the authors propose a new model that is different from the columnar or segmental prosomeric model, they should indicate this in the text but also explain in detail all the features of their new model in relation to why it is not a variation of either the prosomeric or the columnar model. In my opinion, considering how it is presented in the text and the details added so far, the authors propose some changes at specific points in the prosomeric model, but not a new model. In summary, in my opinion, the paper sounds like a claim for a new model, but they provide some sort of variations for the basal rostral and floor plate components of the prosomeric model. The authors should either describe in depth the fundamental basis of the model they are proposing or assume that they are proposing changes to that model. However, they need to include details on why some works showed a complex partition with two rostral neuromeres related to the hypothalamus and one to the prethalamus. How will the authors address this complexity in the adult with this new simplified version that mixes prethalamic and hypothalamic territories? (see below for more details and take into account these articles to think if this simplified version is consistent with the complexity of the adult as can be seen as an example in Ferran et al., 2015-Frontiers for hypothalamus-, Bilbao et al.2022 for the hypothalamus, and Puelles et al., 2021-Jcomp neurology for the prethalamus).

Regarding the concept of hypothalamus, the authors do not provide a definition of what the hypothalamus is and why it is. If they have a model, they need to argue conceptually based on which characteristics the derivatives will be hypothalamic. In addition, there is a problem because the authors mix the prethalamic territory with the telencephalon and the hypothalamus in one unit (according to their explanation in the text). They do not explain if it is an entity like a neuromere and in that case they do not explain how this large segment will be divided in future stages. Furthermore, the authors do not reconcile their proposal with the abundant literature that follows the prosomeric description that provides a complex framework of the simplest ones proposed by the authors. This point needs to be discussed and explained carefully throughout the text (see the articles on hypothalamus related to the prosomeric model included in their bibliography -refs 4 and 6- and the previous one I mentioned).

Regarding the third point, classical fate maps in chickens used quail-chicken grafts (Cobos et al., 2001; Sanchez-Arrones et al., 2009; García-López et al., 2004; Pombero et al., 2009; Alonso et al., 2021) using a precise frame to determine the graft location on the neural plate or neural tube to more accurately correlate territory with fate at later stages. However, the authors used as fate map strategy Dil/DiO injections to determine the way as the forebrain is growing. Usually, this kind of early fate map requires the use of a grid to measure the distance that gives more specificity to the graft in relation to the proposed position of the graft (in that case the Dil/DiO injections) (See Fig. 1 in Pombero and Martinez to know better my question and Fig. 1 in García-López et al., 2004). I couldn't find any information about the use of a coordinate system that warranties the position of the injection that you are assuming in your mapping. Another point is related to assumptions about growth patterns when using a technique that includes relatively large areas of injections (see line 76 in their work). In that case, my impression is that there is no reference to ensure that the injections were carried out in the claimed regions or if the authors interpreted the injection point considering where they observed the mark in later states (Fig S2 is useful to know in same way the position but not with enough detail that will give grid coupled to this figures). These experiments need a grid at the beginning that includes images of the initial injection (at least for validation) so that they can be compared in later stages (see previous comments and papers cited). This precise information is essential to consider the specificity of the proposed fate map. In short, the authors must find a way to demonstrate that they are placing the injection in the location they claim.

Specific comments:

Title: This title does not exactly reflect the results obtained. The study focuses solely on the organization of the rostral forebrain and, above all, on the arrangement of the components of the basal plate and the floor of the hypothalamic and prethalamic regions. If the authors are "solving" anything, it is not about the organization of the forebrain, but rather they are suggesting an alternative option for these parts of the forebrain at some specific stages of development. Furthermore, it gives the impression that the study is a suggestion that will modify some aspects of the prosomeric model, in that sense this work would be improving the prosomeric model or proposing a variant of it. The authors propose an alternative view for the arrangement of mainly rostral neuromeres of the basal plate, as they do not explain in detail whether this is a new model, suggesting that they are proposing changes to the prosomeric model.

Abstract:

Line 26. Authors are following the definition of forebrain including diencephalon proper (p1p3 prosomeres) and secondary prosencephalon (peduncular and terminal neuromeres). However, the fate map is mainly devoted to the secondary prosencephalon, p3, and alar plate of p2, not covering the prosomere 1. Authors fate mapped large region of different forebrain domains

Line 36. According to the data provided by authors they are proposing changes in the prosomeric model of the rostral forebrain mainly in the basal and floor plates. The proposed tripartite model if is matched with the prosomeric ideas that need to be better explained and developed. Some concerns are that these basal plate subdivisions are not continuous

dorsally and all of them necessarily need to be connected with floor plate.

Main text:

-Line 50. These sentences need to be revised because they do not exactly apply to the ideas defined by the prosomeric model (check the literature cited in your bibliography and the previous one I mentioned here that deals with the general principles of the prosomeric model - references 4 and 6, but also Puelles 1995 and other articles mentioned above -). Some confusing ideas may have developed in these sentences, probably due to their excessive simplification (check and rephrase the ideas). The prosomeric model suggests that there are two segments (each segment is defined as a transverse unit with roof, wing, basal, and floor plates) related to the origin of the telencephalon and hypothalamus. Furthermore, the model proposes that the evaginated telencephalon derives from the most dorsal alar plate near the roof plate of the peduncular neuromere (the neuromere in front of p3). This peduncular neuromere according to the model includes an alar portion that is close to the alar-basal boundary (nk2.2 positive) and is related to the origin of hypothalamic derivatives such as the paraventricular nucleus. The basal plate of the peduncular neuromere is related to the origin of hypothalamic derivatives such as the dorsomedian or subthalamic nucleus among others. From the dorsal to the ventral position there are differences between the territories that are related to the distance at which they are located, for example, from the organizers of the roof plate or the floor plate. The most rostral neuromere called the terminal neuromere, includes a portion of the dorsal alar plate that will be the preoptic area (close to the roof plate) and a portion of the alar plate close to the alar-basal limit that will be the origin of the retinal territory (not the eye that involves ectodermal and mesodermal components), supraoptic and suprachiasmatic nucleus among others. In addition, the basal plate of the terminal neuromere will be the origin of the ventromedian, arcuate and mammillary nuclei among others. In summary, the prosomeric model proposes that neuromeres have a clear transverse boundary that should be molecularly identified at some point in early development. Furthermore, the prosomeric model proposes that components, for example, such as the preoptic area (unpaired telencephalon) and the ocular territory, but also the anterior paraventricular area, share the alar plate of the same neuromere, but that does not imply molecular similarity with the components of the basal plate. The great complexity of the hypothalamic region added to the complexity of prosomere 3 must be considered from early stages and a simplified version may give some confusing ideas. Components of the basal plate or alar plate between adjacent neuromeres are expected to share some molecular features because they share ventral and dorsal inductions from the floor or roof plates, respectively. Also, the last sentence of this paragraph is a bit odd because the telencephalon is the most dorsal wing plate near the roof plate and does not necessarily have to be the result of a coordinated pattern with the alar or basal plate of the hypothalamus to be included in the same prosomere. The authors' interesting observation in this paper is related to the behavior of different neuronal clones during neural tube growth. However, the authors do not explain in the text why clonal coordination is required to define neuromeres. Also, the clonal relation doesn't mean that they finally will derive in the same components.

-Line 67. A new model of forebrain development cannot be defined in terms of a tripartite hypothalamus. It does not sound correct that the model of forebrain development is a tripartite hypothalamus. The authors need to review this proposition better. The authors propose a new model (if it really is a new model or a modified version of the prosomeric model) of hypothalamic development or hypothalamic-prethalamic development, at least taking into account the explanations given in the text.

Results:

-Line 78. The authors: "For descriptive purposes, we define the A-P axis as running parallel to the hypothalamic NKX2-2 expression domain and the D-V axis as orthogonal to it (Fig. 1F)". The authors propose a confusing model that has some unresolved problems as a model. For example, they considered a dorsoventral position in scheme 1 F that extends to the rostral tuberal region. The blue line going to the tuberal region runs from dorsal to anterior and not ventrally (unless the authors are considering that there is a rostral floor plate at that point that was not postulated in the text). Furthermore, following this assumption implies that this part of the brain that will be influenced by the prechordal plate will give rise to the floor plate. Does that mean that the alar-basal boundary is in contact with the floor plate? For example, to improve the ideas of this part of the scheme (even if they want to propose a new model), see Ferran et al 2021 (Is there a prechordal region...) for the importance of the prechordal plate and the notochord in establishing the main partitions of the brain bauplan. If the authors want to propose another option, they must consider this conceptual framework which must also be resolved in their model.

-Line 98. The growth lines should be interpreted as representing the patterns of the neuroepithelial clones growing throughout longitudinal domains involving several neuromeres (cases 4-5 for example) and inner portions of the neuromeres as the case of 8-9 (inner pallial and subpallial subdivision that can be visible at clonal levels). In summary, the same information can be used to reinforce prosomeric interpretations. I don't understand or is not conceptually clear defined why the authors consider that the way of patterning growing will be used to redefine the relationship between the territories. The authors need to provide the mechanistic support or evidence to validate this assumption.

-Line 182. This interpretation is not clear. What does it mean bulk of hypothalamus? According to your interpretation some basal plate components but not from the prosomeric model are in a caudal neuromere. Some portions of the tuberal region are ventrally positioned to the telencephalon according to your schema.

-Line 202. Authors "The claimed segmental identity between the telencephalon and hypothalamus implies that fate conversion should be possible only between these identities." According to the prosomeric model, FoxG1 is expressed in the unpaired and evaginated telencephalon (terminal and peduncular neuromeres, respectively). However, the alar hypothalamus (absence of Shh expression) is found in most of the paraventricular territory and subparaventricular territory characterized by the absence of FoxG1. In the animals of this study treated with Cyclopamine it seems as if both FoxG1 and PAX6 were ventralized (See in Ferran et al., 2007 Fig. 1 and 5E, PAX6 expression is also observed at stage HH24 in the paraventricular territory. At early stages an increase in signal can be seen in this territory). In my opinion, these results can

be perfectly interpreted considering the detailed description of the forebrain regionalization supported by the prosomeric model.

-Line 220. This paragraph shows an interesting observation that the authors use to propose a different boundary compared to the prosomeric boundaries. This is an interesting observation but not a definitive demonstration that this distortion represents the actual interprosomeric boundary. The peculiar clonal expansion of the basal plate below the ZLI (assuming that the ZLI is only on the wing plate or if we extend it to the basal plate, it does not necessarily have to follow these lines) needs to be cautiously interpreted and defined precisely if the identity is changes or not on those stages and with the impact in adult brain. There are several patterns like Lim1 (see Ferran et al., 2007 Fig. 6 A that could be interpreted as the interprosomeric boundary between p2/p3). The authors make an interesting observation but do not imply that it represents the boundary between territories. The fate map through later stages with definitive derivatives will provide better support for understanding common subsequent identities to recognize possible derivatives of each neuromere. However, if the authors wish to propose such a boundary, they need to develop a model that indicates the basal plate position for the redefined regions. In addition, more details need to be provided on the components derived of each territory.

-Line 276: The authors propose a change in the interpretation of the pattern observed in HH20 that will help them reinterpret the interprosomeric boundaries. This is an interesting point of view, but not a definitive demonstration of the actual prosomeric interpretations. Of course, it remains an interesting possibility to be considered but requires more detail from the authors on what the model they propose looks like. The authors assume that all bands are specifically related to the proposed derivatives. Molecular maps at this early stage do not necessarily reflect the complete or definitive fate of these territories. Further work examining detailed fate maps will be necessary to demonstrate the true value of early mapping with accurate comparisons of fate maps reaching late stages when definitive derivatives are already found. As mentioned above, some patterns suggest a p2/p3 interprosomeric boundary below ZLI (Lim1), which adds confusion to the resolution of this issue. This distortion observed by the authors could be a behavior of the basal plate near the midline that is then definitively specified and be a longitudinal band.

Figure 1:

For example, Fig. 1 L indicates the injection supposedly in area 11, marking the prethalamus. However, prethalamus region touch the roof plate, and it is worth asking whether the extension to the roof is not representing an expected identification of the fate map reaching the dorsal midline that was not considered in your schemas.

Fig. S3 F. Is green Shh expression?

Discussion:

The discussion should be enriched by aspects such as what are the differences between the proposed model and the prosomeric one (or is it a variation of the prosomeric model?). In addition, a discussion that considers the complexity of the adult brain is necessary because the author proposes a simple model that does not consider all the derivatives proposed in the prosomeric model. Some discussion will also be necessary, for example, on the dorsoventral position of the peduncular fibers in the peduncular prosomere, because with this model some things have become difficult to explain.

Some general considerations need to be made in this regard. It is true that clonal growing is the key to define the boundaries? To what extent do they help or not?

Final comment to consider in your discussion:

Do related clones reflect the identity of the territory at each stage? We see the behavior of related clones, but that does not necessarily mean that they belong to the same neuromere at all stages. If the territory is expanded and exposed to different organizers, it may be differentially affected in different dorsoventral domains. In summary, clonal relatedness suggests possible identities conserved through time, but may not necessarily be the key to defining anteroposterior neuromeric boundaries. In my opinion, future fate maps should be performed on individual cellular components to better understand clonal relationships. Then, at different stages, those patterns should be compared with molecular maps, but later stages with defined derivatives are essential because at early stages molecular maps do not always represent precise boundaries (see figure 14 in Sánchez Arrones et al 2009, for a better idea about the problem).

Version 1:

Reviewer comments:

Reviewer #1

(Remarks to the Author)

I congratulate the authors for vastly improving the manuscript. It is now a very clear and convincing work of great interest for the dev bio community.

The only point I still am not comfortable with is the sole use of HCR results to claim growth pattern distort DV axis in mouse (Figure 4) and would like the authors to make a more careful conclusion from these data, stating that the staining suggests distortion rather than saying it shows that it happens.

Reviewer #3

(Remarks to the Author)

The authors have made considerable improvements to the manuscript, and many of my initial questions have been addressed. The authors have clarified several key points, which have significantly improved the clarity of the manuscript. However, several key points still require further clarification to ensure the work is fully understandable.

As a general comment:

This is an interesting and provocative work that introduces an alternative hypothesis for how the hypothalamic region develops. The core contribution of this work is to present a new way to interpret differential growth in the context of neural tube specification. The authors propose one possible interpretation, but for the moment the evidence didn't discard other options. Differential growth is not a specification map, and even when interpreted as a fate map, the growth patterns need to be analyzed more carefully and in greater detail to support stronger conclusions. Nevertheless, this enormous amount of data are a valuable contribution to the scientific community as they can be reinterpreted to support different conclusions, not only than those proposed by the authors.

In summary the proposal is interesting and provocative, and independently of the lack of maturity at some points (a simple view about the complexity observed in adult brains), give some fresh ideas to have in mind and to be deeply explored in further works.

Some points to be revised.

Title: According to the data provided this work focuses mainly on hypothalamic organization but not on the forebrain in detail. It is not true that hypothalamus is the main problem implying all the forebrain. The forebrain includes telencephalon (impar and evaginated) prethalamus, thalamus and pretektum, but also the midbrain territory according to the finding by Albuixech-Crespo et al. 2017. Most current definitions include the midbrain as part of the forebrain.

I recommend focusing the title on the hypothalamus, as this is the central challenge and main subject of the work. The results of this study have implications for how the forebrain develops. However, the authors did not address the complex issue of how the pallium and subpallium are specified. My recommendation at least is to focus the title on hypothalamic organization. Focusing this title on the hypothalamus will provide greater clarity on the authors' findings, key results, and the challenging data.

Abstract: I think that the problem is the same, the focus of this study is on the hypothalamus not in the entire forebrain. Some little changes need to be made to clarify it.

Line 36: The authors "Our findings challenge the widely accepted prosomere model of forebrain organisation, and we propose an alternative 'tripartite hypothalamus' model."

The findings of the authors doesn't challenge the prosomeric model, the data of this manuscript challenge the prosomeric interpretation of the hypothalamic organization. If the authors provide in future works additional details for other brain regions like telencephalon, midbrain and even the entire hindbrain they will claim that the challenge the prosomeric model... .

The clearest part is the proposed alternative tripartite hypothalamus model. The title and the abstract will benefit from focusing this work on the hypothalamus.

Main text: The authors: "In this scheme, the forebrain consists of five transverse segments – 'prosomeres' – analogous to the neuromeres of the hindbrain and spinal cord."

Recent studies based on the Albuixech-Crespo et al. 2017 describe that the midbrain is part of the forebrain (see a summary of it for clarification and reference about this change in Ferran et al., 2025, see also its Fig. 1 to understand the idea).

Line 69: The authors: "Here we clarify forebrain developmental organisation." The authors propose some point of view based on interesting data about hypothalamic organization. The study is not completed enough to claim that this study is about forebrain organization due that complex regions like subpallium and pallium were not deeply explored. The most controversial aspect in this study goes around the hypothalamic region.

Line 83 The authors "Our work resolves prior misinterpretation and provides a new model for forebrain developmental organisation that we term the 'tripartite hypothalamus' model." The authors are not providing a new model of forebrain organization; they are providing a new model of hypothalamus or a new interpretation about how the hypothalamic and prethalamic territories can be interpreted. Sound so strange that the new model of the forebrain is the tripartite hypothalamus model. Rewrite this because it is confused and not clear. Forebrain is more than the portions discussed by the authors and more than hypothalamus.

Results:

Line 100: The authors: "For descriptive purposes, we define the A-P axis as running parallel to the hypothalamic NKX2-2 expression domain and the D-V axis as orthogonal to it (Fig. 1F)". The expression "descriptive purpose" is confused and not necessary in text. If for the authors, this is the topological A-P axis is enough mentioning directly that this this the A-P axis based on the comparison with Nkx2.2. Recognizing this axis is essential to deep understanding of this study and as consequences in how to understand the data through the text.

Line 150 "The anteriormost diencephalic floor plate arises from zone 3 (Fig. 1W)." The authors need to define diencephalon, because their considering in their model that this is to rostral diencephalon and is different to other views. To better understand the position is a good point to define that this is the anteriormost diencephalic floor plate defined as.... .

Line 162 change topography for topology, because the conserved are the relations with neighboring territories (topological relations).

Line 229: Authors need to explain that they elongate the roof plate close to the alar basal border (See fig. 3E). This detail is important to highlight because it implies that there is a rostral floor plate derived from the prechordal effects. Another curious thing from these schemes is that they appear to follow a segmental partition of the neural tube like a modified prosomeric

model.

Line 255: "We hypothesised that the early proximity of prospective hypothalamic and diencephalic cells results in a shared topological identity, due to similar exposure to fate-determining cues". This is difficult to understand, "shared topological identity"... . The topology is conserved between all territories along neural tube; some models can share the topology but it is not clear what the author is thinking about with share identity. If they are referring to sharing identity of neighbors' territories it is ok, but they don't need to include the word topology. Or explain better the meaning of their idea.

Line 304. "implicating the posterior hypothalamus as part of the ventral diencephalon." Again authors needs to clarify what is the definition for them about diencephalon.

Line 400 "so could not be designated as prosomere M with certainty" Check this description along the text because there no any prosomere M. The mammillary domain is the most ventral domain of the peduncular prosoemer (or hp1). Check your text and explain better the idea.

Line 521 "sondegig"?? Shh?

Supplementary figure 1. E: Telencephalon include pallium and subpallium. Please, modify accordingly. G: Is green Shh gene expression pattern?

Supplementary figure 4B. According to the schema (dorsal view) injections were made in 10 or limit 10/8 (red). However, the green one appears to be in 11 or limit 9/11 not 5.

Thank you for the opportunity to respond to the reviewers' concerns. Our detailed responses are below:

REVIEWER COMMENTS

Reviewer #1

This is a very useful resource for the developmental neuroscience research community. The authors present a set of nice lineage tracing experiments in the chick forebrain. Based on those, 3D computational models (fantastic movies describing likely 3D temporal progression of domains) and restricted set of HCR mRNA detection, the authors refute the current prosomeric model of forebrain organisation, proposing a modified model. Although the work is very carefully done and of general interest for the field, we have difficulties with the lack of focus of the narrative and some lack of evidence for the model proposed (while agreeing on finding the prosomeric model problematic for the anterior and ventral forebrain). Overall, the work as it is presented currently does not support strongly enough the conclusions made by the authors.

Reviewer 1 accepts that the prosomere model is problematic, but requests additional evidence to support our claims for a new model. We thank the reviewer for their very constructive and helpful feedback, which has led us to perform a number of additional experiments and analyses to strengthen our conclusions, including:

- Quantitative analysis of DiI/DiO lineage tracing data. The new analyses fully support our original conclusions, which were based on qualitative analysis of the same data.
- New mouse HCR data, essential for translating our detailed findings from chicken, revealing close similarities between the two species.
- Additional chick HCR analysis of prosomere model boundaries, and brand new scRNA-Seq analysis, highlighting the lack of evidence for a segmental boundary within the hypothalamus.
- New evidence in support of our model, from additional fate conversion experiments and increased tests in cell mixing experiments.
- Limited re-ordering of results, and extensive text changes, for a simpler narrative. Tighter focus on our argument concerning the position and distortion of the forebrain dorsoventral axis.

Specific points to be addressed:

R1 Q1 *The most precious part of the data is the collection of lineage tracing results. Although the stated n for each of the injected forebrain areas described on Figure 1 is high, there is no quantification of the clone size, shape and position at HH20 across embryos injected for each area at HH10. This quantification is essential to assess variability across injections. Figure S2 suggests a fair amount of variability and same embryos are used for two distinct data points, using different left and right positions of injection (eg. Fig. S2 E & F, H & I, J & K).*

We now provide a detailed quantitative analysis of our fate mapping results, describing outcomes in terms of position, shape and relative size. The **new data is shown in Fig 1G-H, AA-AB and Supplementary fig. S5**. In addition to providing new quantification from the twelve HH10 injection zones, we have included data from injections placed at the intersections of these zones, providing higher spatial resolution, totalling 290 examples. (note that as a consequence of showing more data, original Supplementary fig. S2 is now expanded into **new Supplementary figs S2-S4**). These numbers exclude results from injections covering more than two injection zones, and a further approx. 80 patterns resulting from earlier (HH7-9) injections, which presented very similar patterns (some HH9 examples are newly shown in Supplementary fig. S6). Critically, and as shown in **new Supplementary fig. S5**, variability was minimal, and the results confirm our previous conclusions. As pointed out, in some instances, we used the same embryo for multiple injection sites: this provided internal consistency, further supporting the robustness of our interpretation.

R1 Q2 *The last frame of Movie S2 should be used in an improved Figure 2 to visually match the findings with the model of clonal shapes (if the above quantification of clone size, shape and final position confirms the current conclusion per position on the HH10 map).*

Thank you for this suggestion. We have **improved Figure 2** accordingly.

R1 Q3 *The dynamic model for the yellow area in Movie S4 ... is key to the author's model but does not have*

enough lineage tracing evidence for its proposed origin. Specific lineage tracing of this area (with enough n) is needed.

We generated a lot of data for this region, as we focussed our fate mapping efforts on the barely explored ventral forebrain, and apologise that this was not obvious in the previous version. The yellow area (the *SHH*-negative tuberal hypothalamus) closely corresponds to our outcome area H5. As shown in **new Fig 1AA**, the majority of this zone arises from anterior zone 2 (level with zone 5; please refer to zone 2, 1-2 and 2-5 thumbnails in **new Supplementary fig. S5**). This is accurately represented in our fate map (Fig. 2A).

R1 Q4 *The attempt to refute the prosomeric model is not well articulated, offering partial evidence for the proposed alternative. The most important data would be extensive tracing in area 1, 2, 5 and 7. Fig S4D is key and yet is not showing the position of the injected cells at HH10.*

Thank you for encouraging us to provide additional evidence to better refute the prosomere model and build the case for our alternative model. We have performed a number of additional experiments (see detailed responses to questions below), and have substantially re-ordered the manuscript in order to better articulate our findings. In terms of the lineage tracing queried here, we considered 15 injections in area 1, 29 in area 2, 17 in area 5 and 20 in area 7, plus 21 further injections at the intersections of these zones. Lineage tracing of these regions is shown in Fig. 1I,R-T, Supplementary figures S2-S3, and summarised in **new Figure 1AA-AB and new Supplementary fig. S5**. We have updated Supplementary figure S4D (now S6D) to include the injection area, however this specimen was injected at HH8. Equivalent outcomes arise from injections in HH10 zone 7-8 (DiO spot, result is similar to meeting point of T1 and T2 outcome areas, Fig. 1H) and zone 4-5 (DiI spot, result is similar to H4 outcome area, new Supplementary fig. S5).

R1 Q5 *A substantial portion of the argument for the proposed ‘compartmental’ origins of the hypothalamic areas is based on HCR analyses. However, dynamics in gene expression can’t be used as rigorous support for clonal dynamics. This is a big limitation of the manuscript. Higher resolution lineage tracing is needed to back the HCR conclusion up.*

We agree that HCRs do not provide clonal information and wish to clarify what we are suggesting. Primarily, our model revises the forebrain A-P and D-V axis positions. We propose a new boundary region that extends ventral to the ZLI, and propose that the D-V axis in this area then becomes distorted. Importantly, we provide new evidence (**new Fig. 4A-C,E-G**) that indicates that a similar D-V axis distortion at the base of the ZLI occurs in mouse. But we have never suggested a compartment boundary within the tuberal hypothalamus. The defining feature of our model is that it places the anteriormost portion of the hypothalamus ventral to the telencephalon and the remainder of the hypothalamus ventral to two different diencephalic compartments (prethalamus and ZLI), hence ‘tripartite’.

We have been unable to perform high resolution lineage tracing at the boundary ventral to the ZLI, for biological and technical reasons. Lineage restriction is not established in our boundary region at HH10 (growth lines cross the supramammillary hypothalamus), requiring that injections are performed at later stages. This proved impossible: the third ventricle narrows to a cleft where left and right sides are closely apposed, so manoeuvring within this space spreads dye around and causes the needle to snap. Instead, and as detailed in the response to Q6, we have performed a series of novel mixing experiments.

R1 Q6 *The cell mixing experiment (Fig 4J), where cells separate based on their positional identity (anterior hypothalamic vs posterior thalamic), is key to the proposed model and is missing key controls and ambition. It would need i) mixing the posterior hypothalamic cells with their own (A with A in 4J, matching the P with P) and with cells from the proposed anterior hypothalamic compartment, to see if mixing results match their proposed model.*

Thank you for this suggestion; we apologise for having omitted the controls from the first version of the manuscript (despite having done them). As well as including these, we have now performed a much more extensive analysis, testing the ability of cells from seven different forebrain regions to intermingle or separate. The mixing results support our key claim of a ventral boundary region contiguous with the ZLI. Second, the results reveal that PV hypothalamic cells mix with prethalamic, but not telencephalic cells (data shown in **new Fig. 6D-E and new Supplementary figs. S18-S19**).

Together, the new data that we have added in response to Qs1-6 provides substantial new evidence that refutes the prosomere model. As additional evidence, we provide brand new analyses, including new HCRs and

scRNA-Seq analysis, that again highlights the lack of evidence for a segmental boundary within the hypothalamus (**new Fig. 5P-Q, new Supplementary figs. S14-15**).

RI Q7 *The quality of the DiI/DiO images is very low. And the Cre/tdTomato images are also very hard to interpret because of overabundance of positive cells. A better approach could be to inject fluorescently labelled viruses in regions of interest and see lines of cells that way.*

We have improved the quality of the DiI/DiO images. We discuss the Cre/tdTomato images in the response to Q10.

RI Q8 *The point about dorsalization of ventral diencephalon region with cyclopamine is weak, as the markers they use to indicate telencephalon (*Foxg1*) and prethalamus/PV hypothalamic (*Pax6*) are also expressed in the eye. Given that this treatment is known to lead to cyclopic phenotypes, the markers may be present in 'trapped' retinal cells, instead of proving a fate conversion to TC and pre-thalamus. Much more needs to be done to conclude anything from this experiment.*

Thank you for this insightful comment: we agree that we need to exclude the possibility that these are trapped retinal cells. Since our embryos were analysed at too early a stage to detect mature retinal markers (*VSX2* was attempted), we analysed *PAX2*, finding no evidence for expansion (**new Supplementary fig.16B**). In addition we performed three further experiments. First, we analysed lineage at the same time as inhibiting *Shh* signaling. These studies (**new Fig. 6C**) showed that growth patterns are not altered. Second, we performed additional dorsalisation experiments, analysed with a broader set of markers. These reveal an upregulation of the *ARX* in the basal hypothalamus of dorsalised embryos. Over the time period of analysis *ARX* is largely confined to the prethalamus (just overlapping into the basal hypothalamus) - and is not detected in either eye or telencephalon. Its upregulation after dorsalisation fits with previous studies linking hypothalamic to prethalamic development (**new Fig. 6A-B, new Supplementary fig. S16B**). Third, we have performed *in vitro* experiments that likewise reveal an upregulation of *ARX* in explants of prospective prethalamus/hypothalamus (**new Supplementary fig. S17**). Together these experiments support a fate conversion rather than a cell trapping.

RI Q9 *The comparison between chick and mouse is potentially powerful but only basing it on HCR of a few markers introduces too many possible caveats and shouldn't be considered here unless adding clonal evidence.*

We agree that clonal lineage tracing would provide the strongest evidence, but this is beyond the scope of the current study - although we now note that DiI patterns in mouse, similar to some of those we describe in chick, have been published (<https://doi.org/10.1006/dbio.2000.9616>). However, we were extremely keen to better understand the extent to which our studies may translate to the mouse, and so have instead included additional HCR evidence to support our point about possible evolutionary conservation, focussing our narrative specifically on the position and distortion of the dorsoventral axis. **New Figure 4A-C,E-G** strongly supports the contention that growth of the thalamus distorts the ZLI. The ideas raised in this section are important, because they help to explain the arrangement of progenitor regions in mice, and suggest how they may have been misinterpreted in the prosomere model.

Smaller issues:

RI Q10 *They refer to figure 5K in the discussion. That panel doesn't exist in figure 5.*

Thank you for spotting this; all figure references in the new version of the manuscript have been carefully checked.

RI Q11 *Figure S5 is not useful as the Cre clones can't really be informative if not quantified carefully (size and shape) across many embryos.*

We spent months attempting to perform such studies, prior to our submitting the original manuscript. Unfortunately, despite putting in extensive time and effort, it was not possible to provide a detailed quantitative analysis of these samples. This was due to embryo quality (many non-viable), only 50% being positive for the transgene, and because many of the recombined embryos had either too few or too many clones to be useful, even after optimisation of the TAT-Cre recombinase dose and delivery. We performed, but decided to omit, an analysis of clone size 24 hours after treatment, because there were not enough well-separated clones to provide insights beyond the observation that dorsal clones tended to be larger than ventral clones. We felt nonetheless that Figure S5 was worth including as we could readily discern shapes in the 48 hour samples that were consistent with the DiI/DiO patterns.

R1 Q12 *Some of the HCRs are hard to see especially when they've used grey colour on top of the faint blue background.*

Thank you for this feedback. We have improved the quality of the HCR images as requested.

R1 Q13 *The most important focus of the manuscript is the organisation of the hypothalamus. Yet, the results are written in a very complicated fashion, distracting the reader from the most important conclusions. We would advise to reshape the text to facilitate understanding and clarity of thoughts.*

Thank you for this important prompt. We have made a number of changes and rearrangements to the manuscript to improve our narrative. These include:

- Providing extra context in the introduction
- Consolidating all of our work relating to the chicken D-V axis position and distortion within figure 3
- Using figure 4 to explore the same (D-V axis position/distortion) in mice. Our key novel findings are thereby all delivered early on.
- Figure 5 now exclusively deals with our rebuttal of the prosomere model, expanded in scope for robustness
- Relocating the cyclopamine (/FST/sonidigib) and cell mixing experiments to Figure 6, which is now dedicated to developing our new model
- Expanding our discussion, including clarifying our claims about our new model, as per points made above

We very much hope that these changes have improved the narrative, and help the reader to follow our key conclusions. Again, we are grateful for your careful reading of the original manuscript, and very helpful comments.

Reviewer #2

This manuscript describes a comprehensive and detailed fate-mapping of the chicken forebrain at HH10 using fluorescent dye labeling along with spatial expression pattern analysis of multiple marker genes.

The following new knowledge has been generated from this study:

- 1) Generation of detailed fate map of the chick forebrain at HH10.*
- 2) Generation of a 4D model of forebrain growth.*
- 4) Establishing that eye morphogenesis plays a key role in separation of the telencephalon from the prethalamus and the hypothalamus.*
- 3) Proposal of a "tripartite" model for the hypothalamus.*
- 4) Identification of a major forebrain boundary that cuts through the MM hypothalamus and the ZLI.*

This study does not contain any functional characterization that may shed light on the genes and mechanisms involved in the morphogenesis of the forebrain, particularly the hypothalamus. However, it has brought significant clarity to this phenomenon with respect to the growth movements and the changes in the orientation and positioning of the AP and DV axes. In fact this study has challenged some aspects of the existing prosomere model and proposed a new "tripartite" model for the formation of the hypothalamus. Thus significant new knowledge has been add to the field.

The experiments that have been carried out largely support the conclusions made and the data has been analyzed in depth and interpreted correctly.

The methodology used meets the expected standards in the field and sufficient details have been provided in the supplementary materials for reproduction.

Hence this manuscript is suitable for publication without the requirement of any major changes.

This reviewer had no suggestions, and we thank them for their positive remarks.

Reviewer #3

The authors of “Resolving forebrain development organization by analysis of differential growth patterns” proposed a common anterior compartment of the forebrain with telencephalon, hypothalamus and prethalamus. In addition, the authors claim a new model of forebrain development with a tripartite hypothalamus. This is a detailed study that provides extensive data that aims to support their hypothesis. However, after a detailed analysis of the manuscript, some key points need to be resolved by the authors, including the need to better clarify whether they are proposing a new model or suggesting some changes to the prosomeric model, but also other conceptual and interpretive points (see below).

The main novelty of this work is the improvement of fate map data at stages that have not been sufficiently explored or resolved in the chicken. Fate map information is an essential tool to resolve questions about the specific origin of cell populations and addresses questions such as those raised here. These fate map data compared with molecular data provide the authors with the possibility of reaching some conclusions that are interesting, but not definitive and that in some points require a deep understanding and exploration due to the complexity of the adult brain that seems not to be sufficient in the proposed analysis at these early stages (see below for specific details). On of the controversial aspect supported by the authors is their claimed ZLI distortion. Although the work does not definitively demonstrate its proposal, it is an interesting alternative that should be taken into account to definitively resolve this problem in the future.

We thank the reviewer for taking the time to read our manuscript so thoughtfully, and provide positive comments. We are pleased to provide a detailed point-by-point response to the issues raised.

General comments:

I need to highlight three main aspects that the authors need to improve in the manuscript, one related to the model, another related to the conceptual assumption of the hypothalamus and third one the validity of the selected strategy to resolve the fate map and combining it with the molecular map.

The Reviewer raises a number of points under his/her ‘General comments’, which we have broken down into sections Q1-Q8, responding to each in turn.

R3 Q1 *The first point is related to the model. There are two main and somewhat antagonistic models to understand how, starting from the stages of the neural plate and going through the different stages of the neural tube, the brain is parceled out and the main derivatives of each region have emerged. These models are the columnar model and the prosomeric model. Both models are different because, for example, they assume different axial dimensions that determine different dorsoventral relationships along the anteroposterior axis, but in addition, the prosomeric model establishes an anteroposterior segmentation in terms of neuromeres. These differences concern, for example, the relationship of the roof and floor plates to the components of the neural tube (for details of both models, see, for example, in Puelles 1995; A segmental morphological paradigm for understanding vertebrate forebrains). If the authors propose a new model that is different from the columnar or segmental prosomeric model, they should indicate this in the text but also explain in detail all the features of their new model in relation to why it is not a variation of either the prosomeric or the columnar model. In my opinion, considering how it is presented in the text and the details added so far, the authors propose some changes at specific points in the prosomeric model, but not a new model.*

Thank you for this important suggestion. We have made several text additions to better explain our model and how it differs from the prosomere model. In brief, our model differs from the prosomere model in the following major respects:

1. Our proposed topological boundary within the supramammillary hypothalamus, ventral to the ZLI, excludes floor plate from the anterior forebrain. The anterior forebrain therefore does not meet the prosomere definition of a segmented tissue. Further, while our model describes two forebrain transverse subdivisions, we find no evidence that hp1, hp2 or p3 are segments, in the commonly understood sense in developmental biology, such as lineage restriction or differential expression of signalling pathways associated with cell segregation.
2. As well as disagreeing over the topological relations of basal plate structures, our model emphasises the closer relationship of the paraventricular hypothalamus to the prethalamus, as opposed to the telencephalon, placing this region too in the diencephalon. We also relate the temporal side of the

retina to these diencephalic structures, whereas the prosomere model considers the eye an acroterminal (anteriormost) specialisation.

3. The repositioning of most of the hypothalamus within the diencephalon fundamentally undermines the major concept of the secondary prosencephalon.

Please refer to the following added text from the discussion, which follows our conclusions in the results section.

- “We do not propose a ‘hard’ (lineage restricted) boundary between anterior tuberal/telencephalon and more posterior regions, and view these relationships as topological rather than segmental.”
- “Fundamentally, however, we do not consider the forebrain to be segmented as suggested by the prosomere model. And whereas the prosomere model claims a segment boundary between the prethalamus and PV hypothalamus, we group these regions, reflecting the intimate connection of their progenitors and relative separation from telencephalon by the eye field. Indeed, we further speculate that the eye itself has a ‘split’ identity, the anterior (nasal) side relating to the telencephalon, and the posterior (temporal) side relating to the prethalamus and hypothalamus - consistent with the expansion of both telencephalon and prethalamus into the eye field of eyeless Rax mutants.”
- “In conclusion, we forward a ‘tripartite hypothalamus’ model in which the hypothalamus sits ventral to three regions – the telencephalon, prethalamus/PV hypothalamus, and the ZLI (Fig. 6F). The *LFNG*^{-ve} ZLI and posterior hypothalamus form a topological boundary between the anterior *SIX/FEZ*^{+ve} and posterior *IRX*^{+ve} forebrain, with the SM hypothalamus part of a ventral boundary region associated with lineage restriction.”

R3 Q2 *In summary, in my opinion, the paper sounds like a claim for a new model, but they provide some sort of variations for the basal rostral and floor plate components of the prosomeric model. The authors should either describe in depth the fundamental basis of the model they are proposing or assume that they are proposing changes to that model. However, they need to include details on why some works showed a complex partition with two rostral neuromeres related to the hypothalamus and one to the prethalamus. How will the authors address this complexity in the adult with this new simplified version that mixes prethalamic and hypothalamic territories? (see below for more details and take into account these articles to think if this simplified version is consistent with the complexity of the adult as can be seen as an example in Ferran et al., 2015-Frontiers for hypothalamus-, Bilbao et al.2022 for the hypothalamus, and Puelles et al., 2021-Jcomp neurology for the prethalamus).*

Thank you for your comment. As we have now clarified (above), ours is fundamentally a new model, and we elaborate on this in further comments below. Although it will be important to publish a wide-ranging discussion and critique comparing our model with the prosomere and columnar models, we believe a greater level of detail is better suited to a review. We have attempted to balance the competing needs of detail and accessibility, and have taken into account feedback from colleagues, who have advised us that to include extra detail would make understanding difficult for non-expert readers.

However, to respond to your comment, we do not agree that these papers - although valuable - demonstrated the existence of two rostral neuromeres. We have conducted our own analysis of a selection of the genes reported in Ferran et al. 2017 to distinguish hp1 from hp2, failing to discern evidence for the existence of these two compartments. New analysis of large-scale scRNA-Seq from the developing mouse hypothalamus likewise does not support the segregation of terminal and peduncular paraventricular markers, and instead suggests that some reported regional differences in gene expression patterns may be simply attributed to differences in cell maturity (**new Fig. 5P-Q, new Supplementary figures S14-15**). Although purported prosomere boundaries have twice been redrawn (2003, 2012), no definitive evidence for their existence has been provided. Furthermore, in the absence of features generally associated with boundary functions, such as lineage restriction or differential expression of signalling pathways associated with cell segregation, it is debatable whether the language of segments should be employed, rather than that of regional subdivisions (which we favour).

We have added the following text to clarify our views:

- **Introduction:** “Although influential, the prosomere model is controversial, relying heavily for its support on gene expression patterns in the mid-to-late stage embryonic mouse. Classic features of

segment boundaries, such as lineage restriction, have not been identified at any purported prosomere boundaries anterior to the ZLI (the prosomere dorsal p2/p3 boundary), while the locations of prosomere boundaries in the early ventral forebrain are not precisely defined. And while the model incorporates detailed fate-mapping data - which can contribute valuable insight into topological relationships within the developing forebrain - this evidence is largely restricted to the dorsal aspect of the chicken neural tube, leaving ventral and basal regions under-characterized. A further challenge has arisen from recent studies showing that hypothalamic patterning is closely coordinated with prethalamal, rather than telencephalic, patterning (7–11).”

R3 Q3 *How will the authors address this complexity in the adult with this new simplified version that mixes prethalamal and hypothalamic territories? (see below for more details and take into account these articles to think if this simplified version is consistent with the complexity of the adult as can be seen as an example in Ferran et al., 2015-Frontiers for hypothalamus-, Bilbao et al.2022 for the hypothalamus, and Puelles et al., 2021-Jcomp neurology for the prethalamus).*

Our model is mainly concerned with the topological relations of forebrain regions at patterning stages, which is to say their anterior/posterior and dorsal/ventral positions in relation to each other, and this is what our concluding schematic illustrates. By using a simplified diagram we do not mean to imply that the forebrain is not regionalised into complex territories. The existence of neuromeres and their boundaries is not in principle necessary to explain this complex regionalisation, which we suppose arises through established (plus potentially novel) mechanisms such as long-, medium- and short-range signalling from the neural tube itself and the multitude of surrounding tissues. Our scheme therefore raises no special difficulty in explaining later complexity.

Although the study does not deal directly with patterning mechanisms, we do suggest that the intersection of neural and non-neural tissues in our ventral ‘boundary region’ creates an environment rich in cell signalling interactions that may help to explain the complexity of this part of the forebrain. We also refer in our discussion to the chimeric brain hypothesis, which holds that the anterior neural tube (broadly, *Six3/Rax/Fezf*-expressing areas, i.e. corresponding to our anterior forebrain) represents the fusion of two ancestral nervous systems, creating new combinations of co-expressed genes (and therefore functions) and potentially contributing to the extraordinary variety of cell types within the hypothalamus. See also Albuixech-Crespo et al. 2017, whose anterior ‘HyPTh’ segment in amphioxus appears equivalent to our anterior forebrain. Our scheme therefore aligns well with contemporary evo-devo research, whereas it is difficult to see how the prosomere model fits with these recent insights. We think the anterior forebrain is unique in its complexity and its evolutionary and developmental origins, and that if anything, the prosomere contention that the entire nervous system is epichordal, possessing roof and floor plate as per the rest of the nervous system, is an oversimplification.

R3 Q4 *Regarding the concept of hypothalamus, the authors do not provide a definition of what the hypothalamus is and why it is. If they have a model, they need to argue conceptually based on which characteristics the derivatives will be hypothalamic.*

Thank you for this suggestion. We have added the following text to the results section:

“We use a standard definition of the hypothalamus, namely the paraventricular (PV), tuberal, mammillary (MM) and supramammillary (SM) domains, omitting the preoptic region which is now usually considered part of the telencephalic subpallium.”

In the discussion, we now state:

“We therefore accept the subthalamic nucleus as part of the hypothalamus, a view that has gained considerable traction recently.”

R3 Q5 *In addition, there is a problem because the authors mix the prethalamal territory with the telencephalon and the hypothalamus in one unit (according to their explanation in the text). They do not explain if it is an entity like a neuromere and in that case they do not explain how this large segment will be divided in future stages.*

Our study is not concerned with the mechanisms of forebrain patterning, which fall outside the scope of the work. Please refer to our above answers on the subjects of forebrain patterning (Q3) and the clarification of our proposed model (Q1).

R3 Q6 *Furthermore, the authors do not reconcile their proposal with the abundant literature that follows the prosomeric description that provides a complex framework of the simplest ones proposed by the authors. This*

point needs to be discussed and explained carefully throughout the text (see the articles on hypothalamus related to the prosomeric model included in their bibliography -refs 4 and 6- and the previous one I mentioned).

The prosomere literature references (and incorporates) many excellent and impressive studies, but we disagree with some of the reasoning and conclusions presented therein. Similar to our comment above, we believe that a review is required to do justice to this subject, and that to provide this level of discussion within a primary research article unfortunately would distract from the most important messages and compromise readability.

R3 Q7 *Regarding the third point, classical fate maps in chickens used quail-chicken grafts (Cobos et al., 2001; Sanchez-Arrones et al., 2009; García-López et al., 2004; Pombero et al, 2009; Alonso et al., 2021) using a precise frame to determine the graft location on the neural plate or neural tube to more accurately correlate territory with fate at later stages. However, the authors used as fate map strategy Dil/DiO injections to determine the way as the forebrain is growing. Usually, this kind of early fate map requires the use of a grid to measure the distance that gives more specificity to the graft in relation to the proposed position of the graft (in that case the Dil/DiO injections) (See Fig. 1 in Pombero and Martinez to know better my question and Fig. 1 in García-López et al., 2004). I couldn't find any information about the use of a coordinate system that warrants the position of the injection that you are assuming in your mapping.*

Thank you for this comment. We greatly admire these studies, and their conclusions have accurately predicted our Dil fate mapping results in dorsal regions. Reflecting this, we performed far fewer injections in dorsal regions than were required for the ventral neural tube (please refer to **new Supplementary figure S5B**). However, in our own, earlier work (Fu et al., 2017) we found that using a distance-based coordinate system led to inconsistent results for the ventral neural tube, and we therefore developed a morphology/landmark-based system. In the present study we extended this approach, using a grid based on morphological criteria (see Figure 1D). Regardless of the respective merits of the two different approaches, you will see from our new quantitative analysis (**new Supplementary figure S5C**) that the strategy we employed led to consistent results - and we consider this to validate our method. Please also see supplementary fig. S1C, which provides examples of how we confirmed the accuracy of our targeting at HH10.

R3 Q8 *Another point is related to assumptions about growth patterns when using a technique that includes relatively large areas of injections (see line 76 in their work). In that case, my impression is that there is no reference to ensure that the injections were carried out in the claimed regions or if the authors interpreted the injection point considering where they observed the mark in later states (Fig S2 is useful to know in same way the position but not with enough detail that will give grid coupled to this figures). These experiments need a grid at the beginning that includes images of the initial injection (at least for validation) so that they can be compared in later stages (see previous comments and papers cited). This precise information is essential to consider the specificity of the proposed fate map. In short, the authors must find a way to demonstrate that they are placing the injection in the location they claim.*

We are happy to clarify our workflow for this process. The injection position for every sample was recorded individually in a lab notebook and a photograph was also taken for every example (a number of examples are presented in **new Supplementary figures S2-S4** - expanded from original figure S2). Both were referred to when interpreting the results later on, because the photographs carry only partial (2D) information that is insufficient to distinguish. The experimenter who performed the injections designated the position with respect to the twelve zones in Figure 1D, and these were independently scored by two others. Concordance was extremely high and the few disagreements (approximately 5% of injections) were settled by discussion including reference back to lab notes.

Specific comments

The Reviewer raises a number of points under his/her 'Specific comments', which we have broken down into queries Q9-Q25, responding to each in turn.

R3 Q9 *Title: This title does not exactly reflect the results obtained. The study focuses solely on the organization of the rostral forebrain and, above all, on the arrangement of the components of the basal plate and the floor of the hypothalamic and prethalamic regions. If the authors are "solving" anything, it is not about the organization of the forebrain, but rather they are suggesting an alternative option for these parts of the forebrain at some specific stages of development. Furthermore, it gives the impression that the study is a suggestion that will modify some aspects of the prosomeric model, in that sense this work would be improving the prosomeric model or proposing a variant of it. The authors propose an alternative view for the arrangement*

of mainly rostral neuromeres of the basal plate, as they do not explain in detail whether this is a new model, suggesting that they are proposing changes to the prosomeric model.

Although we indeed focus most heavily on the position of the rostral forebrain, we do present DiI data as far back as the pretectum, and we interpret our results in a whole-forebrain context. We therefore consider it reasonable to use 'forebrain' in the title. As we have clarified above, our model is novel and not a variant of the prosomere model.

Abstract:

R3 Q10 Line 26. Authors are following the definition of forebrain including diencephalon proper (p1p3 prosomeres) and secondary prosencephalon (peduncular and terminal neuromeres). However, the fate map is mainly devoted to the secondary prosencephalon, p3, and alar plate of p2, not covering the prosomere 1. Authors fate mapped large region of different forebrain domains

As above, we do have some injections that labelled prospective pretectum (e.g. see zone 12 results, **new Supplementary fig. S5C**) so we think it a fair approximation to describe the work as fate mapping the forebrain.

R3 Q11 Line 36. According to the data provided by authors they are proposing changes in the prosomeric model of the rostral forebrain mainly in the basal and floor plates. The proposed tripartite model if is matched with the prosomeric ideas that need to be better explained and developed. Some concerns are that these basal plate subdivisions are not continuous dorsally and all of them necessarily need to be connected with floor plate.

As we have explained above, our model is not a version of the prosomere model and therefore we are not bound to restrictions based on the prosomere definition of segments.

Main text:

R3 Q12 -Line 50. These sentences need to be revised because they do not exactly apply to the ideas defined by the prosomeric model (check the literature cited in your bibliography and the previous one I mentioned here that deals with the general principles of the prosomeric model - references 4 and 6, but also Puelles 1995 and other articles mentioned above -). Some confusing ideas may have developed in these sentences, probably due to their excessive simplification (check and rephrase the ideas). The prosomeric model suggests that there are two segments (each segment is defined as a transverse unit with roof, wing, basal, and floor plates) related to the origin of the telencephalon and hypothalamus. Furthermore, the model proposes that the evaginated telencephalon derives from the most dorsal alar plate near the roof plate of the peduncular neuromere (the neuromere in front of p3). This peduncular neuromere according to the model includes an alar portion that is close to the alar-basal boundary (nk2.2 positive) and is related to the origin of hypothalamic derivatives such as the paraventricular nucleus. The basal plate of the peduncular neuromere is related to the origin of hypothalamic derivatives such as the dorsomedian or subthalamic nucleus among others. From the dorsal to the ventral position there are differences between the territories that are related to the distance at which they are located, for example, from the organizers of the roof plate or the floor plate. The most rostral neuromere called the terminal neuromere, includes a portion of the dorsal alar plate that will be the preoptic area (close to the roof plate) and a portion of the alar plate close to the alar-basal limit that will be the origin of the retinal territory (not the eye that involves ectodermal and mesodermal components), supraoptic and suprachiasmatic nucleus among others. In addition, the basal plate of the terminal neuromere will be the origin of the ventromedian, arcuate and mammillary nuclei among others. In summary, the prosomeric model proposes that neuromeres have a clear transverse boundary that should be molecularly identified at some point in early development. Furthermore, the prosomeric model proposes that components, for example, such as the preoptic area (unpaired telencephalon) and the ocular territory, but also the anterior paraventricular area, share the alar plate of the same neuromere, but that does not imply molecular similarity with the components of the basal plate. The great complexity of the hypothalamic region added to the complexity of prosomere 3 must be considered from early stages and a simplified version may give some confusing ideas. Components of the basal plate or alar plate between adjacent neuromeres are expected to share some molecular features because they share ventral and dorsal inductions from the floor or roof plates, respectively. Also, the last sentence of this paragraph is a bit odd because the telencephalon is the most dorsal wing plate near the roof plate and does not necessarily have to be the result of a coordinated pattern with the alar or basal plate of the hypothalamus to be included in the same prosomere.

We have updated our text to include more detail on the prosomere model, making clear that we are referring to two segments, for example:

Introduction:

“Each prosomere possesses both roof plate and floor plate, a conceptual necessity to meet the prosomere definition of a segment. The hypothalamus is grouped with the telencephalon, together occupying two anteriormost prosomeres (hp2, hp1). Posterior to these are three diencephalic prosomeres (p3-p1) housing the prethalamus (p3), thalamus (p2), preteetum (p1) and associated ventral structures (4, 6).”

As our results are complex, we have been compelled to present only those core aspects of the prosomere model that are necessary for the understanding of our manuscript.

R3 Q13 *The authors' interesting observation in this paper is related to the behavior of different neuronal clones during neural tube growth. However, the authors do not explain in the text why clonal coordination is required to define neuromeres. Also, the clonal relation doesn't mean that they finally will derive in the same components.*

Thank you for this comment. We do not claim that clonal patterns define regions in and of themselves, but this information should guide our interpretation of the relationships between regions because cells that arise close together experience similar patterning signals, which determine regional identity during early developmental stages. This point is strongly backed up by the results of our dorsalisation experiments (**new Fig. 6A-C**; Supplementary fig. S16), which indicate that early spatial origin predicts competence to express regional markers.

We have added the following comments to the discussion:

“Growth patterns help us to understand the epigenetic history and environmental exposures of cells during a crucial period of specification, relevant for both stem cell differentiation efforts and topological models. Our results reveal early proximity of much of the developing basal hypothalamus to the diencephalic prethalamus and thalamus; confirm a close association of prospective prethalamus and PV hypothalamus; and suggest that telencephalic and basal hypothalamic lineages are largely segregated. We conducted our fate mapping through patterning stages, when cell fate is initially plastic, but becomes progressively restricted according to the positional information encountered. Thus, although it must be stressed that growth lines do not directly delineate forebrain regions, the many similarities in the shapes of DiI growth lines and gene expression domains demonstrate the close coupling of growth and fate.”

We have also improved our explanation of our results by adding the following statement to the corresponding results section:

“Considered alongside our fate map (Fig. 2A, Movie S1), these results suggest that proximity to prospective telencephalon at HH10 predicts competence to express *FOXG1*, and proximity to prospective prethalamus/PV hypothalamus predicts competence to express *PAX6/ARX*.”

Our suggestions are further backed up by cell mixing experiments, including new data presented in **new Figure 6D-E**. Hence, our conclusions are based on several different lines of evidence.

R3 Q14 *-Line 67. A new model of forebrain development cannot be defined in terms of a tripartite hypothalamus. It does not sound correct that the model of forebrain development is a tripartite hypothalamus. The authors need to review this proposition better. The authors propose a new model (if it really is a new model or a modified version of the prosomeric model) of hypothalamic development or hypothalamic-prethalamal development, at least taking into account the explanations given in the text.*

Thank you for this suggestion. As we highlight in the Introduction “the fundamental layout of the developing vertebrate brain is still disputed, reflected in a confused and contradictory contemporary literature, with the extent and position of the hypothalamus at the centre of the debate”, therefore our placement of the hypothalamus clearly differentiates our model from both columnar and prosomere models – something we want to convey in the name. Importantly, the name we have chosen refers not to the hypothalamus in isolation, but to its position *with respect to other structures*. These topological relations are inseparable from the orientations of the A-P and D-V axes, so considering that we have placed the hypothalamus ventral to three major regions, the tripartite hypothalamus and its dorsal relations together constitute the majority of the forebrain, their described relations revealing – simply but clearly – the axis directions across the same area.

We have added the following clarification to the discussion:

“In conclusion, we forward a ‘tripartite hypothalamus’ model in which the basal hypothalamus sits ventral to three regions – the telencephalon, prethalamus/PV hypothalamus, and the ZLI (Fig. 6F). The *LFNG*^{-ve} ZLI and posterior hypothalamus form a topological boundary between the anterior *SIX/FEZ*^{+ve} and posterior *IRX*^{+ve} forebrain, with the SM hypothalamus part of a ventral boundary region associated with lineage restriction.”

Results:

R3 Q15-Line 78. *The authors: “For descriptive purposes, we define the A-P axis as running parallel to the hypothalamic NKX2-2 expression domain and the D-V axis as orthogonal to it (Fig. 1F)”. The authors propose a confusing model that has some unresolved problems as a model. For example, they considered a dorsoventral position in scheme 1 F that extends to the rostral tuberal region. The blue line going to the tuberal region runs from dorsal to anterior and not ventrally (unless the authors are considering that there is a rostral floor plate at that point that was not postulated in the text). Furthermore, following this assumption implies that this part of the brain that will be influenced by the prechordal plate will give rise to the floor plate. Does that mean that the alar-basal boundary is in contact with the floor plate? For example, to improve the ideas of this part of the scheme (even if they want to propose a new model), see Ferran et al 2021 (Is there a prechordal region...) for the importance of the prechordal plate and the notochord in establishing the main partitions of the brain bauplan. If the authors want to propose another option, they must consider this conceptual framework which must also be resolved in their model.*

We included Figure 1F only to show the reader how to interpret the directional terms in our results section, although it does fit with our final scheme (please refer to Figure 6F). These comments appear to follow from the assumption that we have provided a prosomere update rather than a new model, which we have now clarified. Accordingly, by definition the blue line does not run from dorsal to anterior because this aligns with the proposed D-V axis in our model. Likewise, we are not suggesting that a floor plate exists in the anterior tuberal region, because the anterior forebrain in our model does not possess a floor plate in the usual sense of the word. On the question of how closely the ventral midline (rather than the Arx^{+ve} floor plate) approaches the alar-basal boundary, this is an interesting unresolved question, and indeed, in principle the two could meet. We have added the following comment to the discussion:

“We do not define a specific point in the anterior tuberal hypothalamus where the ventral midline ends anteriorly, but note that in principle it could extend far enough to abut the ABB.”

Regarding the prechordal plate and notochord, our discussion includes the comments:

“We speculate that the SM hypothalamus and flanking $WNT8B$ -positive regions represent an extensive ventral ‘boundary region’ (Fig. 5K), where the juxtaposition of anterior ($FEZF^{+ve}$) and posterior (IRX^{+ve}) forebrain - plus underlying tissues such as notochord and prechordal mesoderm - provided an evolutionary substrate for the generation of high signalling and cell type complexity.”

“The rest of the hypothalamic midline lacks floor plate markers at mid-embryogenesis but arises from $Foxa1^{+ive}$ Shh^{+ive} floor plate-like (HypFP) progenitors (12, 50), its uniqueness accounted for by the distinct evolutionarily origin of the anterior forebrain (51–54), and mechanistically by the distinct patterning influence of the prechordal mesoderm (55–57).”

R3 Q16-Line 98. *The growth lines should be interpreted as representing the patterns of the neuroepithelial clones growing throughout longitudinal domains involving several neuromeres (cases 4-5 for example) and inner portions of the neuromeres as the case of 8-9 (inner pallial and subpallial subdivision that can be visible at clonal levels). In summary, the same information can be used to reinforce prosomeric interpretations. I don’t understand or is not conceptually clear defined why the authors consider that the way of patterning growing will be used to redefine the relationship between the territories. The authors need to provide the mechanistic support or evidence to validate this assumption.*

Please see our response to Q13 where we have outlined our thoughts on the relevance of the growth patterns to interpreting the relationships between regions, including comments on new evidence presented (**new Fig. 6D-E**). Regarding the prosomere model, as we describe in our manuscript (Figure 3A-C, associated text and references therein), our work disproves the growth model invoked in support of the prosomere model. Alternative interpretations are welcome, but as we have suggested in various comments above (e.g. Q2), we do not believe the prosomere model is well supported and believe a fresh proposition is warranted.

R3 Q17-Line 182. *This interpretation is not clear. What does it mean bulk of hypothalamus? According to your interpretation some basal plate components but not from the prosomeric model are in a caudal neuromere. Some portions of the tuberal region are ventrally positioned to the telencephalon according to your schema.*

Thank you for requesting greater precision. We now refer specifically to the **ventral** hypothalamus and state that the anterior tuberal hypothalamus originates ventral to the telencephalon:

“Instead, our fate-mapping shows that much of the **ventral** hypothalamus arises posterior to the telencephalon, and ventral to areas that will give rise to the PV hypothalamus, prethalamus and thalamus (Fig. 2A,F; Fig. 3D-E). The anterior tuberal hypothalamus is an exception, originating ventral to the telencephalon.”

R3 Q18-Line 202. Authors “The claimed segmental identity between the telencephalon and hypothalamus implies that fate conversion should be possible only between these identities.” According to the prosomeric model, *FoxG1* is expressed in the unpaired and evaginated telencephalon (terminal and peduncular neuromeres, respectively). However, the alar hypothalamus (absence of *Shh* expression) is found in most of the paraventricular territory and subparaventricular territory characterized by the absence of *FoxG1*. In the animals of this study treated with Cyclopamine it seems as if both *FoxG1* and *PAX6* were ventralized (See in Ferran et al., 2007 Fig. 1 and 5E, *PAX6* expression is also observed at stage HH24 in the paraventricular territory. At early stages an increase in signal can be seen in this territory). In my opinion, these results can be perfectly interpreted considering the detailed description of the forebrain regionalization supported by the prosomeric model.

Thank you for these thoughtful comments. In fact we have now rearranged our manuscript (for the sake of clarity), and the cyclopamine analysis is presented without reference to the prosomere model, as we have already presented our reasons for proposing an alternative.

R3 Q19 -Line 220. This paragraph shows an interesting observation that the authors use to propose a different boundary compared to the prosomeric boundaries. This is an interesting observation but not a definitive demonstration that this distortion represents the actual interprosomeric boundary. The peculiar clonal expansion of the basal plate below the ZLI (assuming that the ZLI is only on the wing plate or if we extend it to the basal plate, it does not necessarily have to follow these lines) needs to be cautiously interpreted and defined precisely if the identity is changes or not on those stages and with the impact in adult brain. There are several patterns like *Lim1* (see Ferran et al., 2007 Fig. 6 A that could be interpreted as the interprosomeric boundary between p2/p3). The authors make an interesting observation but do not imply that it represents the boundary between territories. The fate map through later stages with definitive derivatives will provide better support for understanding common subsequent identities to recognize possible derivatives of each neuromere. However, if the authors wish to propose such a boundary, they need to develop a model that indicates the basal plate position for the redefined regions. In addition, more details need to be provided on the components derived of each territory.

Our model is a first iteration, but one we believe is well-supported by our data. We agree that further work will be necessary to elucidate the precise nature of our proposed boundary, but this would be beyond the scope of the current study. We would like to highlight the numerous genes that we have analysed in the posterior hypothalamus, all of whose profiles support our model (thank you for mentioning the interesting *Lim1* expression pattern, which appears to be excluded from our boundary region at its posterior edge). In contrast, more than thirty years since its first iteration (Puelles & Rubenstein 1993), no such description exists of prosomere boundaries in the basal plate at stages equivalent to chicken HH20 (approximately mouse E11). We show that the prosomere boundaries as described in mouse cannot be resolved correctly in the HH20 chicken embryo (Fig. 5H-L), or adequately in the E7 chicken (Fig. 5M-O).

Although we agree that it would be interesting to analyse fate mapping results at later stages, we chose to focus on a relatively early endpoint (HH20) for two reasons. First, this time-point marks the end of patterning stages, certainly for the hypothalamus. Species conservation is high at this time point, and regional marker genes can then be used to identify the spatial location of cells in single cell datasets for example (including mouse, human), whose lineages may then be reconstructed. We have added the following text to the discussion: “All major progenitor zones are established in the chicken hypothalamus by the experimental endpoints used, enabling the inference of adult nuclei origins by comparison to longer term fate mapping studies and developmental single cell sequencing datasets, including to some extent cross-species data. Continued rapid progress in understanding the ventral forebrain in particular should confirm these inferences, directly or indirectly.”

Regarding the SM hypothalamus specifically, we draw the readers’ attention to the derivatives of this progenitor zone, as it occupies a key position in our model. From the discussion:

“Cells originating in the SM region contribute widely to the posterior hypothalamus and beyond, with *PITX2/LMX1B*^{+ve} progenitors in mouse contributing to the SM, subthalamic and ventral premammillary nuclei”

Secondly, we chose to end our studies at an early stage to avoid the complication of the large tangential migrations that occur within the forebrain. These movements do not necessarily help to elucidate developmental organisation, because lineage restriction (for example between rhombomeres) generally occurs in the ventricular zone but often not the mantle. We believe there is a limit to the usefulness of extrapolating from later to earlier stages, and consider the detailed focus on early stages a major strength of this study.

R3 Q20-Line 276: *The authors propose a change in the interpretation of the pattern observed in HH20 that will help them reinterpret the interprosomeric boundaries. This is an interesting point of view, but not a definitive demonstration of the actual prosomeric interpretations. Of course, it remains an interesting possibility to be considered but requires more detail from the authors on what the model they propose looks like. The authors assume that all bands are specifically related to the proposed derivatives. Molecular maps at this early stage do not necessarily reflect the complete or definitive fate of these territories. Further work examining detailed fate maps will be necessary to demonstrate the true value of early mapping with accurate comparisons of fate maps reaching late stages when definitive derivatives are already found. As mentioned above, some patterns suggest a p2/p3 interprosomeric boundary below ZLI (Lim1), which adds confusion to the resolution of this issue. This distortion observed by the authors could be a behavior of the basal plate near the midline that is then definitively specified and be a longitudinal band.*

Please see our response to questions 1 (extra detail on our model) and 19 (discussion on boundaries and stages analysed).

R3 Q21 Figure 1:

For example, Fig. 1 L indicates the injection supposedly in area 11, marking the prethalamus. However, prethalamic region touch the roof plate, and it is worth asking whether the extension to the roof is not representing an expected identification of the fate map reaching the dorsal midline that was not considered in your schemas.

Thank you for pointing this out - our original text was confusing as we did not distinguish between the prethalamus itself (which arises in zone 11) and the prethalamic eminence. We have now labelled these areas separately in Fig. 1E, and provide text and images (**new fig. S2F-G**) describing the origin of the prethalamic eminence in posterior zone 9:

“Injection zones 7-9 give rise to growth lines that reproducibly stretch from the telencephalon to the dorsal optic stalk or optic midline (Fig. 1I-L; fig. S2) - zone 9 also skirting the prethalamic eminence (fig. S2).”

R3 Q22 Fig. S3 F. Is green Shh expression?

Thank you, this has been fixed (now fig. S2K).

R3 Q23 Discussion:

The discussion should be enriched by aspects such as what are the differences between the proposed model and the prosomeric one (or is it a variation of the prosomeric model?). In addition, a discussion that considers the complexity of the adult brain is necessary because the author proposes a simple model that does not consider all the derivatives proposed in the prosomeric model. Some discussion will also be necessary, for example, on the dorsoventral position of the peduncular fibers in the peduncular prosomere, because with this model some things have become difficult to explain.

Thank you for this suggestion. We have improved our discussion on the difference between the two models (please see e.g. question 1). From the discussion:

“The defining feature of our model is that it places the hypothalamus ventral to the telencephalon *and* to two parts of the diencephalon (prethalamus and ZLI), hence ‘tripartite’. Therefore, the majority of the hypothalamus is considered ventral diencephalon, similar to classical columnar views. However, we have adopted the curved ABB from the prosomere model, and link the anterior tuberal hypothalamus and telencephalon. We also note that earlier versions of the prosomere model placed axes more similarly to ours. Therefore, our model incorporates elements from both columnar and prosomere models.”

Our model aims to describe early *developmental* organisation in terms of axes and major forebrain divisions, rather than to outline in detail the derivatives of the early hypothalamic (and other) progenitor zones. Similar to our comment for question 13, having worked out the HH10 fate map to ~HH18, existing work can be used to broadly infer later outcomes, and rapid progress is being made in this area.

Regarding the peduncular fibres, similar to our response to Q3, we do not accept that segment boundaries are required to explain the existence of complex patterned territories. In the absence of specific evidence for segment boundaries at early developmental stages, we suppose that ordinary patterning mechanisms are responsible for these phenomena. We include the following note in the results section:

“While *Meis2* expression did taper anteriorly within the SPa (fig. S15E), it is unclear why this - or indeed any of the above genes showing graded differential expression - should indicate a segmental boundary, rather than being the result of any other patterning process.”

R3 Q24 *Some general considerations need to be made in this regard. It is true that clonal growing is the key to define the boundaries? To what extent do they help or not?*

Please see our response to question 13.

R3 Q25 *Final comment to consider in your discussion:*

Do related clones reflect the identity of the territory at each stage? We see the behavior of related clones, but that does not necessarily mean that they belong to the same neuromere at all stages. If the territory is expanded and exposed to different organizers, it may be differentially affected in different dorsoventral domains. In summary, clonal relatedness suggests possible identities conserved through time, but may not necessarily be the key to defining anteroposterior neuromeric boundaries. In my opinion, future fate maps should be performed on individual cellular components to better understand clonal relationships. Then, at different stages, those patterns should be compared with molecular maps, but later stages with defined derivatives are essential because at early stages molecular maps do not always represent precise boundaries (see figure 14 in Sánchez Arrones et al 2009, for a better idea about the problem).

We agree that a range of evidence must be considered when approaching questions of organisation. Please see our earlier comments (question 13 on the relevance of growth patterns, and question 19 on boundaries and stages considered). We appreciate the point about the imprecision of using early stages, but the Sánchez Arrones paper performed forebrain fate mapping at HH4 and analysed at HH8. This is a very different proposition to the HH17-20 forebrain (= our main endpoint), in which patterning is largely complete and neurogenesis well underway, having begun at approximately HH11. Forebrain cell clones labelled at HH18 tend to stay close together rather than dispersing widely (e.g. Larsen 2001).

REVIEWER COMMENTS

Reviewer #1 (Remarks to the Author):

I congratulate the authors for vastly improving the manuscript. It is now a very clear and convincing work of great interest for the dev bio community.

We thank the reviewer for this comment and their many helpful suggestions.

The only point I still am not comfortable with is the sole use of HCR results to claim growth pattern distort DV axis in mouse (Figure 4) and would like the authors to make a more careful conclusion from these data, stating that the staining suggests distortion rather than saying it shows that it happens.

We agree that this point should have been made cautiously. We have made the following changes.

Line 302: subheading changed to "A possible dorsoventral axis distortion in mouse"

Line 314: figure heading changed to "Growth patterns may distort the dorsoventral axis in mouse"

Lines 325-326: "Expansion of the thalamus may displace the Shh^{ve} ZLI, causing the angle between the ZLI and posterior hypothalamic Shh expression domain to become more acute ..."

Lines 332: "To explore whether this may point towards an axis distortion as seen in chicken ..."

Lines 340-343: "... thalamic growth may contribute to this change by displacing the dorsal ZLI anteriorly. Although it is not possible to resolve this question solely by analysing gene expression patterns, we propose that the D-V axis runs through the ZLI and SM hypothalamus and may become distorted in mice as in chicken."

Reviewer #3 (Remarks to the Author):

The authors have made considerable improvements to the manuscript, and many of my initial questions have been addressed. The authors have clarified several key points, which have significantly improved the clarity of the manuscript. However, several key points still require further clarification to ensure the work is fully understandable.

We thank the reviewer for their comments, and their helpful suggestions for improving the manuscript

As a general comment:

This is an interesting and provocative work that introduces an alternative hypothesis for how the hypothalamic region develops. The core contribution of this work is to present a new way to interpret differential growth in the context of neural tube specification. The authors propose one possible interpretation, but for the moment the evidence didn't discard other options. Differential growth is not a specification map, and even when interpreted as a fate map, the growth patterns need to be analyzed more carefully and in greater detail to support stronger conclusions. Nevertheless, this enormous amount of data are a valuable contribution to the scientific community as they can be reinterpreted to support different conclusions, not only than those proposed by the authors.

We have been clear on this point: lines 587-590: "Thus, although it must be stressed that growth lines do not directly delineate forebrain regions, the many similarities in the shapes of Dil growth lines and gene expression domains demonstrate the close coupling of growth and fate."

In summary the proposal is interesting and provocative, and independently of the lack of maturity at some points (a simple view about the complexity observed in adult brains), give some fresh ideas to have in mind and to be deeply explored in further works.

Some points to be revised.

Title: According to the data provided this work focuses mainly on hypothalamic organization but not on the forebrain in detail. It is not true that hypothalamus is the main problem implying all the forebrain. The forebrain includes telencephalon (impar and evaginated) prethalamus, thalamus and pretectum, but also the midbrain territory according to the finding by Albuixech-Crespo et al. 2017. Most current definitions include the midbrain as part of the forebrain.

I recommend focusing the title on the hypothalamus, as this is the central challenge and main subject of the work. The results of this study have implications for how the forebrain develops. However, the

authors did not address the complex issue of how the pallium and subpallium are specified. My recommendation at least is to focus the title on hypothalamic organization. Focusing this title on the hypothalamus will provide greater clarity on the authors' findings, key results, and the challenging data.

Respectfully, we disagree that our data focuses mainly on hypothalamic organisation, and that the title should be altered to reflect this. Both our fate-mapping and our cell mixing studies cover the telencephalon, eyes, prethalamus, thalamus, pretectum, as well as the hypothalamus. The question of how the pallium and subpallium are specified is far beyond the scope of this manuscript. Altering the title would minimize the importance of our study, and would not reflect its wider contextual importance.

Abstract: I think that the problem is the same, the focus of this study is on the hypothalamus not in the entire forebrain. Some little changes need to be made to clarify it.

We accept that we did not study the midbrain, which proponents of the prosomere model argue to be the posterior forebrain. However, this is not yet common usage. To emphasise that our study is relevant to the entire forebrain (our italics, below), we have made slight rearrangements to the abstract:

Lines 32-39: “Comparative gene expression analysis and cell mixing experiments suggest the existence of a contiguous transverse boundary region, encompassing the zona limitans intrathalamica (ZLI) and retromammillary hypothalamus, that divides the anterior and posterior forebrain, and becomes distorted at the ZLI base. Fate conversion experiments indicate that the hypothalamus is topologically tripartite, lying ventral to the telencephalon, prethalamus and ZLI. Our findings challenge the widely accepted prosomere model of forebrain organisation, do not support a segmented anterior forebrain, and instead suggest a ‘tripartite hypothalamus’ model.”

Line 36: The authors “Our findings challenge the widely accepted prosomere model of forebrain organisation, and we propose an alternative ‘tripartite hypothalamus’ model.”

The findings of the authors doesn’t challenge the prosomeric model, the data of this manuscript challenge the prosomeric interpretation of the hypothalamic organization. If the authors provide in future works additional details for other brain regions like telencephalon, midbrain and even the entire hindbrain they will claim that the challenge the prosomeric model.

The clearest part is the proposed alternative tripartite hypothalamus model. The title and the abstract will benefit from focusing this work on the hypothalamus.

Line 37-38: Again, we respectfully disagree. Our model is fundamentally incompatible with the prosomere model as it directly challenges the existence of the secondary prosencephalon and the existence of segments anterior to the ZLI, i.e. p3, hp1 and hp2 prosomeres. Likewise it directly challenges the orientation of the axes. Instead, our model substantially expands the diencephalon by including most of the hypothalamus and the temporal retina.

Main text: The authors: “In this scheme, the forebrain consists of five transverse segments – ‘prosomeres’ – analogous to the neuromeres of the hindbrain and spinal cord.”

Recent studies based on the Albuixcetch-Crespo et al. 2017 describe that the midbrain is part of the forebrain (see a summary of it for clarification and reference about this change in Ferran et al., 2025, see also its Fig. 1 to understand the idea).

Lines 51-58: we have updated the text to reflect this: “In this scheme, the forebrain consists of seven transverse segments – ‘prosomeres’ ... Most recently the midbrain has recently been designated as forebrain, therefore contributing two more prosomeres.”

Line 69: The authors: “Here we clarify forebrain developmental organisation.” The authors propose some point of view based on interesting data about hypothalamic organization. The study is not completed enough to claim that this study is about forebrain organization due that complex regions like subpallium and pallium were not deeply explored. The most controversial aspect in this study goes around the hypothalamic region.

Please see our response to requests with regard to the title and line 36.

Line 83 The authors “Our work resolves prior misinterpretation and provides a new model for forebrain developmental organisation that we term the ‘tripartite hypothalamus’ model.” The authors are not providing a new model of forebrain organization; they are providing a new model of hypothalamus or a new interpretation about how the hypothalamic and prethalamic territories can be interpreted. Sound so strange that the new model of the forebrain is the tripartite hypothalamus model. Rewrite this because it is confused and not clear. Forebrain is more than the portions discussed by the authors and more than hypothalamus.

Please see our response to requests with regard to the title and line 36.

Results:

Line 100: The authors: “For descriptive purposes, we define the A-P axis as running parallel to the hypothalamic NKX2-2 expression domain and the D-V axis as orthogonal to it (Fig. 1F)”. The expression “descriptive purpose” is confused and not necessary in text. If for the authors, this is the topological A-P axis is enough mentioning directly that this is the A-P axis based on the comparison with Nkx2.2. Recognizing this axis is essential to deep understanding of this study and as consequences in how to understand the data through the text.

Line 103: we have removed the expression “For descriptive purposes”

Line 150 “The anteriormost diencephalic floor plate arises from zone 3 (Fig. 1W).” The authors need to define diencephalon, because their considering in their model that this is to rostral diencephalon and is different to other views. To better understand the position is a good point to define that this is the anteriormost diencephalic floor plate defined as....

Thank you for pointing this out. We have altered the wording to read ‘The anteriormost SHH+ve floor plate arises from zone 3’. See also our response to your comment about line 304 regarding the diencephalon.

Line 162 change topography for topology, because the conserved are the relations with neighboring territories (topological relations).

Thank you, this has been corrected

Line 229: Authors need to explain that they elongate the roof plate close to the alar basal border (See fig. 3E). This detail is important to highlight because it implies that there is a rostral floor plate derived from the prechordal effects. Another curious thing from these schemes is that they appear to follow a segmental partition of the neural tube like a modified prosomeric model.

We have added a legend to panel E in Figure 3 to clarify the meaning of the orange and green zones, which indeed represent the roof plate and floor plate in A, but in E they represent the dorsal and ventral midlines. We have added new text to the figure legend to clarify the meaning of the dotted lines, which are intended to illustrate the growth pattern, not to indicate segments:

Lines 240-247: “**A** ... Transverse lines represent incipient interprosomeric boundaries. ... **E** ... Dotted lines represent cells at particular transverse positions at the earlier stage, and their descendants’ positions after neural tube bending.”

Line 255: “We hypothesised that the early proximity of prospective hypothalamic and diencephalic cells results in a shared topological identity, due to similar exposure to fate-determining cues”. This is difficult to understand, “shared topological identity”... The topology is conserved between all territories along neural tube; some models can share the topology but it is not clear what the author is thinking about with share identity. If they are referring to sharing identity of neighbors’ territories it is ok, but they don’t need to include the word topology. Or explain better the meaning of their idea.

Lines 261: We have improved the phrasing, to read: “a similar positional identity”

Line 304. “implicating the posterior hypothalamus as part of the ventral diencephalon.” Again authors needs to clarify what is the definition for them about diencephalon.

We have clarified this in lines 308-310: “As the ZLI is intercalated between the prethalamus and thalamus – both diencephalic regions – this implicates the posterior hypothalamus as part of the ventral diencephalon.”

See also lines 547-549: “Similar to the classical view, we therefore tentatively group the PV hypothalamus with the prethalamus as part of the diencephalon, posterior to the telencephalon.”

Lines 638-640 clarify this further: “the ZLI and ventral boundary region cut through the diencephalon, dividing the forebrain into anterior and posterior parts.”

Line 400 “so could not be designated as prosomere M with certainty” Check this description along the text because there no any prosomere M. The mammillary domain is the most ventral domain of the peduncular prosoemer (or hp1). Check your text and explain better the idea.

Line 400-402: We now refer back to the relevant information: “(21,32), a *SIM1/NKX2-1*^{ve}, *FOXA1/OTP*^{ve} M domain (Fig. 5C,D) could not be identified in the HH20 chicken (Fig. 5L).

Line 407-408: This now reads: “and so could not be designated as the prosomere model M domain with certainty”.

Line 521 “sondegig”?? Shh?

We have clarified this in Lines 488-490: “We therefore dorsalisated the chicken hypothalamus by applying SHH inhibitors (cyclopamine or Sonidegib)”

Supplementary figure 1. E: Telencephalon include pallium and subpallium. Please, modify accordingly. G: Is green Shh gene expression pattern?

Thank you for spotting these omissions, which have been corrected.

Supplementary figure 4B. According to the schema (dorsal view) injections were made in 10 or limit 10/8 (red). However, the green one appears to be in 11 or limit 9/11 not 5.

Thank you for spotting this - it has been corrected